# Influence of the previous North Atlantic Oscillation (NAO) on the spring dust aerosols over North China

**Yan Li[1], Falei Xu[1], Juan Feng[2], Mengying Du[1], Wenjun Song[1], Chao Li[1,3], and Wenjing Zhao[1,4]**

[1]Key Laboratory for Semi-Arid Climate Change of the Ministry of Education, College of Atmospheric Sciences, Lanzhou University, Lanzhou, China

[2]College of Global Change and Earth System Science, Beijing Normal University, Beijing, China

[3]Hubei Key Laboratory for Heavy Rain Monitoring and Warning Research, Institute of Heavy Rain, China Meteorological Administration, Wuhan, China

[4]Gansu Meterological Service Center, Lanzhou, China

**Correspondence:** Juan Feng (fengjuan@bnu.edu.cn)

**Abstract.** The North Atlantic Oscillation (NAO) has been confirmed to be closely related to the weather-climate in many regions of the Northern Hemisphere, however, its effect and mechanism upon the formation of regional dust events (DEs) have rarely been involved in China. By using the station observation data, and multi reanalysis datasets, the influence of NAO on the dust aerosols (DAs) in China, as well as the corresponding mechanism of synoptic cause are explored in perspective of transient eddy fluxes. It is found that the DAs in the non-dust source areas show high values with a strong annual variability in north of the Yangtze River (30-40°N, 105-120°E), which is referred to as the North China hereafter. A significant negative relationship is indicated between the boreal winter NAO index and the late spring DAs in the North China with the correlation coefficient of -0.39. According to the 9 spring DEs affected significantly by negative phase of the preceding winter NAO in the North China during 1980-2020, it is shown that before the outbreak of DEs, due to the transient eddy momentum (heat) convergence (divergence) over the dust aerosol (DA) source regions, the zonal wind speed increases in upper-level troposphere, strengthening the zonal wind in the middle-lower levels through momentum downward transmission. Simultaneously, there is transient eddy momentum (heat) divergence (convergence) around the Ural Mountains, which is favorable for establishment and maintenance of the Ural ridge, as well as development of the air temperature and vorticity advections. The combined action of temperature and vorticity

advections results in the Siberian Highs and Mongolian cyclone to establish, strengthen, and move southward near the surface, guiding the cold air from high latitudes southward, and is favorable to the uplift and transmission of DAs to the downstream North China. After the outbreak of DEs, change of transient eddy fluxes in the DA source regions and the Urals regions, leads to both energy and mass divergence and reduction of the zonal winds over the North China. Accompanying with the prevailing southerly airflow in south of the North China, a stable high value of DA concentration is maintained for 1-2 days. This study reveals the impact of transient eddy fluxes transport on the dusty weather anomalies modulated by the NAO negative signal in the North China, which can deepen the understanding of formation mechanism of DEs in China.

## 1 Introduction

Airborne dust aerosols (DAs), which play a significant role in the evolution of the atmospheric system, have an important influence on human society, ecosystems, and biochemical cycles. In particular, the radiative forcing of DAs is comparable to that of clouds on a regional-scale and has a key impact on the local weather-climate (Kaufman et al., 2002; Huang et al., 2014a). Recently, the influence of DAs on the weather-climate has attracted widespread attention from researchers, and the direct, indirect and semi-direct feedback responses between them have been confirmed by scientific studies. For direct processes, DAs can impact the weather-climate by scattering shortwave and longwave radiation to cool the atmosphere or by absorbing longwave radiation and solar shortwave radiation released by the earth and clouds to heat the atmosphere (Sokolik and Toon, 1996; Zhang et al., 2019); via indirect influence, DAs can change the local rainfall processes as well as the albedo of ice and snow surfaces by altering the characteristics of cloud condensation nuclei and ice nuclei (Ackerman et al., 2000; Sassen et al., 2003); and in the semi-direct effects, DAs can increase cloud droplet evaporation and reduce cloud water pathways by heating the clouds (Huang et al., 2006). In addition, it has been found that due to the increase of DAs in the Arabian Sea, Arabian Peninsula, and West Asia, the Indian summer monsoon will be enhanced and thus influence the rainfall in central India (Jin et al., 2016). The DAs of worldwide annual emissions in the range of 1000-2150 Tg are lifted up by the surface wind (Zender et al., 2004), with only approximately 30% of these resettles in the dust aerosol (DA) source regions, while the other 70% is transported

to downstream areas thousands of kilometres away, causing changes in the local weather-climate (Duce et al., 1980; Huang et al., 2015).

One of the most serious natural catastrophes in East Asia is dusty weather. Studying the climatological features, variations, and influencing factors of DAs are crucial to both understanding and predicting DA variation patterns (Feng et al., 2020; Yao et al., 2021). Extensive analysis of meteorological and climatic factors associated with the occurrence of global dust events (DEs) have been conducted using ground-based observations (e.g., Ji and Fan, 2019; Liu et al., 2020), satellite remote sensing (e.g., Chiapello et al., 2005; Han et al., 2022), and numerical simulations (e.g., Ginoux et al., 2004; Chen et al., 2017). It is concluded that the variations in DA concentration and transport are related to many factors, including atmospheric circulation (Huang et al., 2021), surface wind (Liu et al., 2004; Wang et al., 2018), cyclone frequency (Yu et al., 2019), Asian monsoon (Wilcox et al., 2020), and the North Atlantic Oscillation (NAO) (Mao et al., 2011; Li et al., 2019a; Feng et al., 2019). The deserts in northern and western China and Mongolia are the most important areas affecting China (primarily Badain Juran and Taklimakan), and the DAs in the above source areas contribute 70% of the dust emissions in East Asia (Zhang et al., 2003; Che et al., 2019). Therefore, it is of great significance to explore the characteristics and causes of DAs.

According to previous studies, spring is the most active time for the occurrence of DEs in China, accounting for over 80% of all DEs yearly (Liu et al., 2004). According to statistical analysis, the annual dust emissions in northern China are approximately 25 Tg, of which spring emissions account for over half (Xuan et al., 2000). Recently, DEs seem to be frequent in China, seriously endangering human health, hindering socioeconomic development and causing widespread concern among the scientific community and the public. For example, from 14-16 March 2021, a large-scale dust event (DE) occurred in northern China (Yin et al., 2021), resulting in more than 3.8 million $km^2$ affected, with the maximum hourly average concentration of $PM_{10}$ surpassing 8000 um $m^{-3}$ and a drop in visibility to less than 0.5 km in Beijing (Zhang et al., 2022). In addition, approximately 800 Tg of DAs in China are emitted into the atmosphere every year, nearly half of the global emissions (Zhang et al., 1997). Therefore, it is of important practical and scientific value and relevance to investigate the formation mechanism of DEs in China, especially in spring.

In the Northern Hemisphere (NH), the NAO is a seesaw mode of centre pressure variation

(Walker, 1924) near the Azores and Iceland, and is the most important low-frequency dipole pattern during boreal winter. The NAO has crucial effects on temperature (e.g., Hurrell, 1995; Yu et al., 2016), precipitation (e.g., Hartley and Keables, 1998; Giannini et al., 2000), and storm tracks (e.g., Lau and Nath, 1991; Jin et al., 2006) in the North Atlantic and its surrounding areas. However, its signal can also be used as a mediator waveguide through the midlatitude westerly wind belts, capturing the downstream propagating Rossby wave train and thus extending its effect to the weather-climate of the Eurasian continent, as well as the entire NH (e.g., Watanabe, 2004; Lin et al., 2011; Zhang et al., 2021). China is located downstream of the NAO-connected circulation system, and its climate is impacted by variations in different phases and intensities of the NAO. For example, when the NAO is unusually strong in winter, China will be controlled by an abnormally warm and rainy climate during the same period, and the temperature in the Yangtze-Huaihe River Basin will be abnormally cooler in the next summer (Wang and Shi, 2001). Liu et al. (2001) found that changes in the NAO can alter the temporal and spatial distributions of precipitation in the eastern Tibetan Plateau in summer by affecting the tropospheric zonal winds. The NAO also has a key influence on weather-scale anomalies in China. Li et al. (2021a) explored the connection between the NAO and persistent haze events in Beijing and found that the NAO negative signal can modulate wave train transmission to northern China, exacerbating the haze duration in Beijing. Yao et al. (2022) indicated that under the influence of seasonal accumulation of the NAO negative signal, the polar vortex will become more vulnerable and unstable compared to the climatic mean value, which will lead to more extreme cold occurrences in China.

Meanwhile, it has been shown that the NAO also has a crucial influence on the process of DEs in China. For example, Tang et al. (2005) noted that the frequency of spring DEs in northern China was significantly influenced by fluctuations in the NAO intensity. In addition, Zhao et al. (2012) discovered an obvious negative correlation between the winter NAO intensity and the late spring and summer DEs in northwestern China. The above studies mainly analysed the association between the NAO and the DEs in China from the perspective of seasonal-scale climate. However, the influence of weather-scale meteorological elements on DEs is also important. For example, Qian et al. (2002) explored the connection between Chinese DEs and cyclonic activities from 1948 to 1999 and found a highly positive correlation between them. Wang et al. (2009) further discovered that the

spring DEs in northern China were mostly near the centre of the Mongolian Cyclone (MC). Hara et al. (2006) used a regional-scale DA transport model and found that the frequency of DEs in the Gobi Desert of East Asia can also be increased by the intrusion of polar cold air. Furthermore, An et al. (2018) discovered that the decline in the strength and frequency of DEs in East Asia during the period of 2007-2016 was highly associated with the weakening tendency of strong winds following the entrance of higher-latitude cold air. As mentioned previously, the large-scale climate variability model of the NAO can capture the Rossby wave train propagating downstream, which in turn has a crucial effect on the weather-scale elements. As one of the two fundamental fluctuations in the atmosphere, transient eddy is widely used in studies to diagnose Rossby wave trains (Trenberth, 1986) and can provide a perspective of mechanism exploration in abnormal variations in atmospheric circulation (Li et al., 2022). It has been noted that transient eddy has a sustained impact on the development of atmospheric systems such as the Siberian Highs (SH) as well as the North Pacific low. (Holopainen and Oort, 1981; Holopainen et al., 1982). The forcing impact of transient eddy on the mean airflow can enhance anticyclonic circulation, which in turn may trigger large-scale severe low-temperature occurrences (Li et al., 2019b). In addition, Li et al. (2022) showed that transient eddy played a significant role in the formation of abnormal atmospheric circulation of DEs by focusing on the spring DEs in south Xinjiang, China, during 1980-2018.

It is essential to continue deep study of the synoptic mechanism of DE formation through the aspect of the NAO. On the one hand, with the recent huge energy consumption and astounding economic development in East Asia, the eastern part of China has suffered from escalating air pollution problems (Zhang et al., 2012; Zhao et al., 2016). With the occurrence of DEs, DAs are one of the most crucial elements affecting air quality in East Asia (Huang et al., 2014b; Nie et al., 2015). Previous studies have provided some analysis and initial progress on the relationship between the NAO and the DEs in northern China regions, mainly DA source areas such as northwestern China. However, such studies are limited to the eastern part of China, which is not a DA source area but is severely affected by DEs. On the other hand, previous studies of the NAO on the DEs in China have mainly been analysed on the seasonal-scale to provide a large-scale climatic background for the occurrence of DEs in China, but it is not clear how the NAO affects the DEs in China at the synoptic-scale, and the role of transient eddy in the anomalous circulation of the atmosphere in DEs under

the modulation of the NAO is uncertain.

From the above points of view, we investigated three main scientific questions in this paper: 1) Does the NAO affect the area with high values of DAs in eastern China? What are the characteristics of the impact? 2) What are the synoptic causes of the formation of DEs influenced by the NAO? 3) How can we explain the mechanism of the formation of synoptic system anomalies by transient eddy under NAO modulation? To address the above issues, by using station observation data, the MERRA-2 dataset, and reanalysis datasets from 1980-2020, this paper investigated the long-term changes in the spring DAs in China to examine the characteristics of the impact of NAO on the DAs in eastern China during the previous period and the same period, and to explore the atmospheric circulation mechanisms affecting DEs considering transient eddy flux transport under the influence of the NAO. The remaining work is organized as follows. Section 2 describes the datasets and methods employed in this paper. Sections 3.1, 3.2, and 3.3 cover the selection of DA study area and its link with the NAO, the process of abnormal atmospheric circulation during DEs, and the impacts of the transport features of transient eddy fluxes accompanying synoptic system anomalies, respectively. Section 4 contains the conclusions and discussions.

## 2 Datasets and methods

### 2.1 Datasets

The China National Meteorological Center (CNMC) provides the daily DE occurrence dataset over mainland China, which contains three types of DEs (dust storm, blowing dust, and floating dust). When DAs are transported with visibility less than 1 km, it is considered a dust storm, whereas floating dust with visibility less than 10 km is caused by DAs from upwind source regions, and blowing dust is defined similarly to floating dust, with the difference that the DAs are emitted from local source areas. Several previous studies have confirmed the validity of the dataset (Kang et al., 2016; Wang et al., 2018).

Datasets of daily and monthly DA concentrations under the Modern-Era Retrospective Analysis for Research and Applications, version 2 (MERRA-2), were derived from the Global Modeling and Assimilation Office (GMAO) of the National Aeronautics and Space Administration (NASA) (Gelaro et al., 2017). Its foundation is built on the assimilation of multiple satellite systems

(AVHRR, MISR, MODIS) and AERONET ground-based observations, and the correctness and reliability are considered to be better than those obtained from the assimilation of individual satellites. The MERRA-2 DA dataset is used for the Goddard Earth Observing System Model, version 5 (GEO-5) (Molod et al., 2015) and the grid statistical interpolation analysis method for three-dimensional variable assimilation (3DVar) (Kleist et al., 2009). The most significant benefit of the MERRA-2 dataset is the temporal and spatial coherence, which cannot be surpassed by station-based or individual satellite data and allow for rigorous statistical analysis of the temporal and spatial patterns (Gelaro et al., 2017). The dust column mass density was used to represent the atmospheric DA concentration under the MERRA-2 product during the 1980-2020 period (horizontal resolution: 0.625° × 0.5°) in this paper. Yao et al. (2021) used the MERRA-2 dataset to analyse the monthly mean DAs in China and found that the DAs from March to July showed higher values with obvious variabilities, which was consistent with the previous conclusions obtained using different DA datasets (Che et al., 2019; Liu et al., 2020), demonstrating the feasibility and applicability of the MERRA-2 dataset for assessing the DAs in China.

The Sea Surface Temperature (SST) dataset was obtained from the Extended Reconstructed Sea Surface Temperature, version 5 (ERSST-5), from the National Oceanic and Atmospheric Administration (NOAA) for the period 1980-2020 (horizontal resolution: 2.0° × 2.0°) (Huang et al., 2017). The atmospheric reanalysis datasets, including the wind field, geopotential height field, sea level pressure field, temperature field, and vertical velocity field, were obtained from the European Center for Medium-Range Weather Forecasts, version 5 (ERA-5), over the period 1980-2020 (horizontal resolution: 0.25°x 0.25°). Compared to its predecessor, ERA-Interim, ERA5 has a modified data assimilation system and improved physical model to achieve reanalysis data information with improved quality.

In this paper, unless otherwise specified, the boreal winter season is referred to as December-January-February (DJF), and the spring season is March-April-May (MAM).

### 2.2 Methods

The NAO index (NAOI) was chosen to indicate the NAO activities. The NAOI describes the large-scale circulation characteristics of the NAO well (Li and Wang, 2003), and is defined by the

following equation:

$$NAOI = \hat{P}_{35^\circ N} - \hat{P}_{65^\circ N} \quad (1)$$

In the above equation, $P$ represents the monthly mean sea level pressure averaged from 80°W to 30°E, $\hat{P}$ is the standardized value of $P$, and the subscript $\hat{P}$ indicates the latitude. The selection criteria for the NAO abnormal years are based on the NAOI index averaged over the winter months, the index is then normalized, and the years with a NAOI exceeding 0.5 standard deviations are recorded as NAO anomalous years.

According to the operational criteria of the CNMC, DE occurrences are defined as when the number of sites with DEs is more than 1/3 of the total amount in the selected region. According to the variations in DAs and to assure the feasibility of DE sample number, we defined a DE when any of three types of DEs occur. Although the DEs last for a short period of time, even only one day, the abnormal atmospheric circulation that affects the DEs will exist over a long period before and after the day of the outbreak of DEs (Li et al., 2022). Therefore, to investigate the abnormal atmospheric signals during DEs, the time-scale was separated into the following parts: “Day -n”, “Day 0” and “Day +n”, which indicate previous to, the date of, and lag after the day of DE outbreak, respectively. To probe obvious signals in DEs, we extended the number of days of DE cycle to 6 days, i.e., before the DE outbreak to “Day -3” and after the DE outbreak to “Day +2”.

To analyse the features of transient eddy flux transport during DEs based on atmospheric variables, the method of physical decomposition was used in this work (Qian, 2012). Any atmospheric element $F$, such as *u*, *v*, and *T*, can be decomposed into two parts according to the above decomposition principle: the temporal mean part $\overline{F}$ and the transient eddy part $F'$. $\overline{F}$ represents the state in which the radiation from the subsurface at a fixed point in the atmosphere is in equilibrium with the daily and annual cycles of solar radiation, while $F'$ is a deviation from the balanced state as follows:

$$F = \overline{F} + F' \quad (2)$$

In this paper, the transient eddy flux transport is denoted by $[u'v']$ and $[v'T']$, representing the momentum and heat of transient eddy transport, respectively. Within the range of longitude that we choose, the zonal means are represented in the variables by square brackets $([\ ])$.

## 3 Results

### 3.1 Selection of the dust aerosol study area and association with the NAO

The concentration of spring DA and their standard deviation distributions in China are shown in Fig. 1. The large values of DAs are mostly situated in source regions such as Xinjiang and Inner Mongolia. In addition, the DA concentrations in the non-dust source areas north of the Yangtze River (30-40°N, 105-120°E) also show high values (Fig. 1a); that is, the spring DAs show higher concentrations here than in other regions of China. Moreover, the standard deviation of DAs in this region is also characterized as a large value, indicating that DAs have strong annual variability and are easily influenced by dust disasters (Fig. 1b). Therefore, the area spanning (30-40°N, 105-120°E) is selected as the study region to explore the variability of DAs and the possible formation causes of DEs, which in the analysis that follows is referred to as North China.

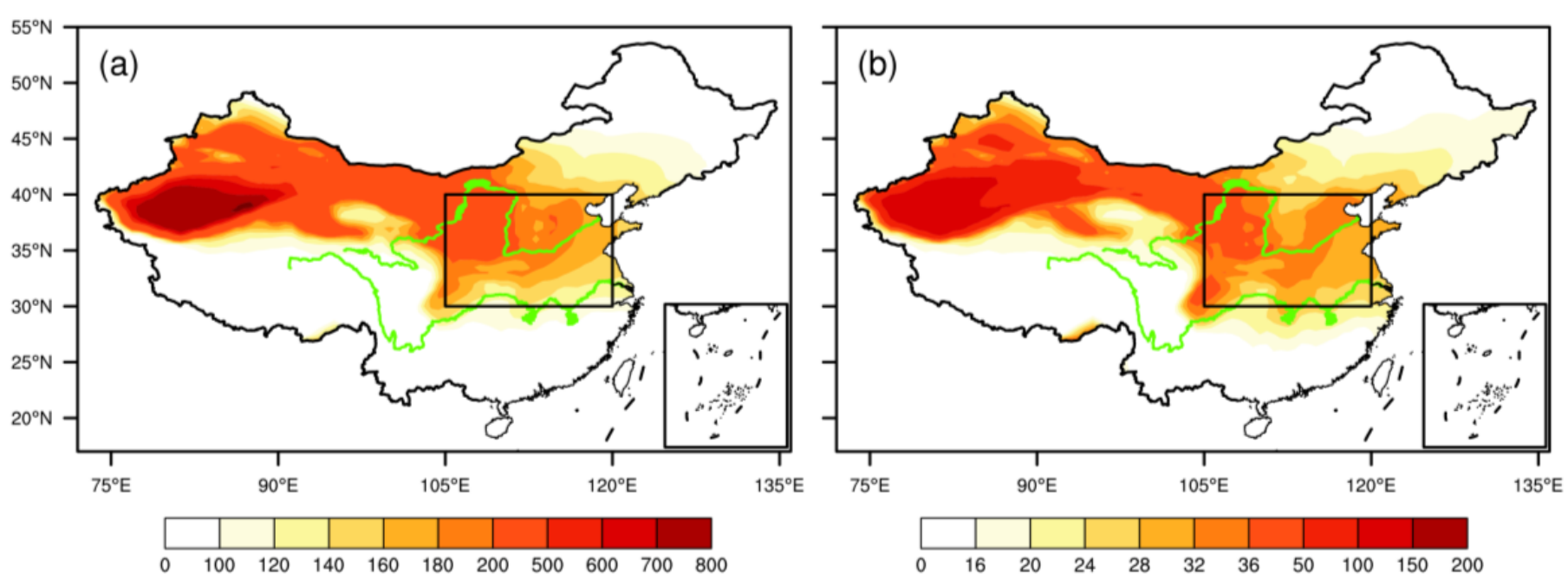

Figure 1. Spatial distribution of the climatologically averaged dust column mass density (a) and dust column mass density standard deviation (b) in spring during 1980-2020, unit: mg $m^{-2}$. The black box indicates the North China study area.

To explore the possible links affecting the spring DAs in North China, the correlation coefficients between the regional average DAs and the geopotential height field are illustrated in the previous and contemporaneous periods (Fig. 2). The pattern of the MAM DAs and the DJF, JFM, FMA and MAM geopotential height fields all show significant north–south reversal in the North Atlantic, i.e., negative in the Azores and positive in Iceland, which indicates typical characteristics of the NAO negative phase. This dipole structure can be observed in the lower, middle and upper

troposphere, denoting that there is an important connection between the spring DAs in North China and the previous NAO variations. Furthermore, significant correlation coefficients can be found between the spring DAs in North China and the NAOI in DJF, JFM, and FMA in the early period, and the correlation coefficients are all approximately -0.40. Simultaneously, as illustrated in Fig. 3, March, April and May, corresponding to the spring months, show substantial standard deviations, suggesting that the DAs in North China vary dramatically in these 3 months, while the monthly average standard deviations of the NAOI present the highest values in December, January and February, corresponding to the winter. Therefore, the effect of the NAO in boreal winter on the later spring DAs in North China and the synoptic formation mechanism of the impact should be analysed and explored. Considering the significant negative effect of the NAO on the DAs in North China, the main focus in the subsequent analysis is on the negative phase of the NAO.

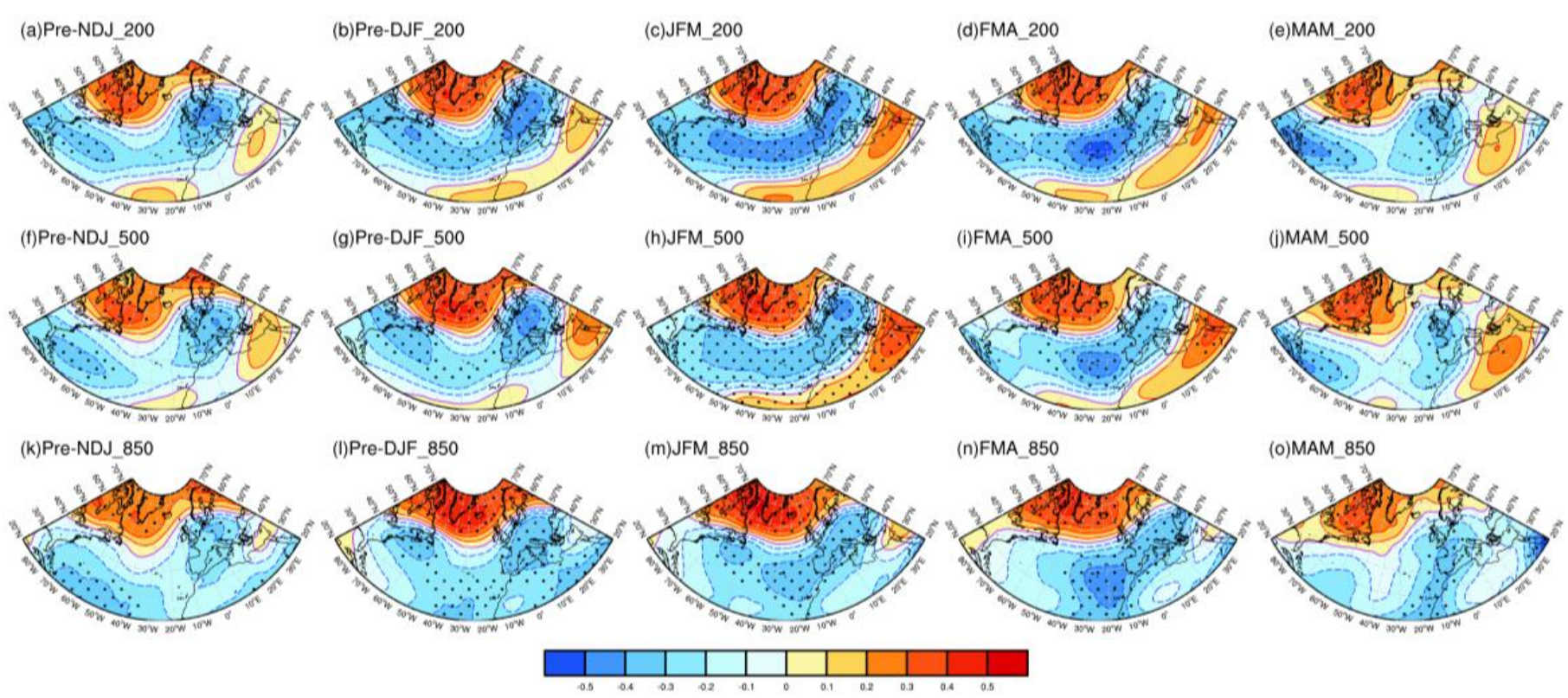

Figure 2. Correlations of the spring dust column mass density in North China from 1980-2020 with the 200 hPa geopotential height anomalies during the previous period and the same period (a-e). (f-j) and (k-o) Same as in (a-e) but for the 500 hPa and 850 hPa geopotential height anomalies. Dotted blue, solid magenta and solid red lines indicate negative, zero and positive contour values, respectively. All contour intervals are 0.1. The black dotted areas are statistically significant at/above the 90% confidence level.

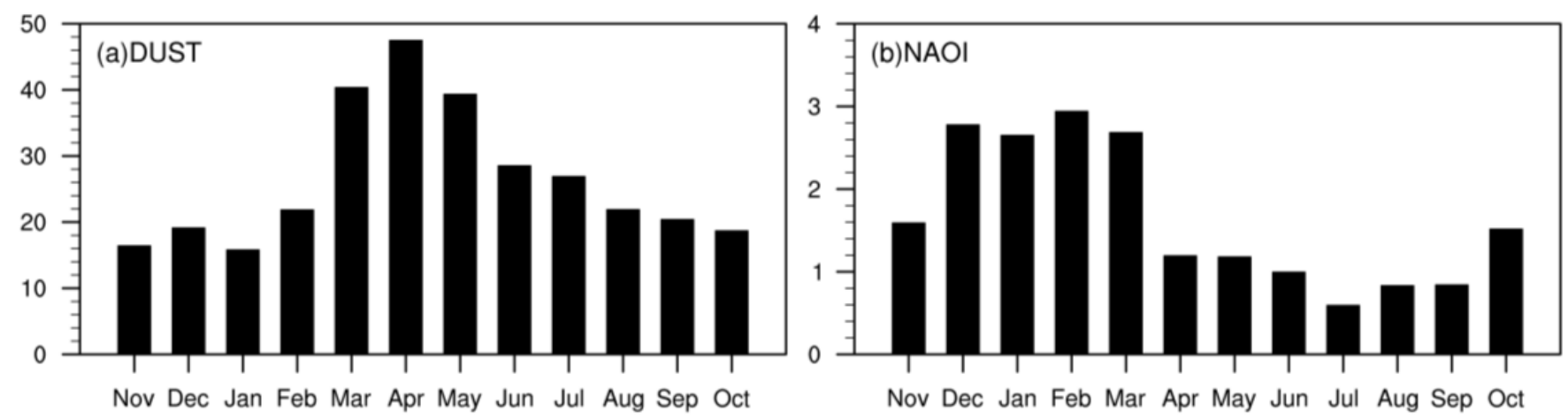

Figure 3. The monthly standard deviation of the dust column mass density in North China (a) and the NAOI (b) during 1980-2020.

According to the analysis, the NAOI in boreal winter and late spring DAs in North China have a substantial negative link. How does the boreal winter NAO affect the late spring DAs in North China? There is an obvious correlation between the boreal winter NAOI and both contemporaneous and late North Atlantic SST anomalies (SSTA), as well as a triple-pole model of "-, +, -" (Fig. 4a to b). Previous studies have noted that on seasonal and annual scales, the triple-pole of SST is the primary mode of North Atlantic SST variation, and its variability is closely correlated to the changes in the NAO (Wu et al., 2009). The NAO can modify SST by influencing changes in sea surface wind speed and hence latent heat fluxes in the North Atlantic (Cayan et al., 1992). Simultaneously, the triple-pole SST mode generates atmospheric circulation similar to the NAO, suggesting a positive feedback between them, which has been verified in both observational data analysis (Czaja et al., 2002) and ocean-atmosphere coupled model simulations (Watanabe et al., 2000). These findings show that the SST has a "capacitor effect" on the NAO negative signal, which prolongs the influence of the NAO signal over the surrounding and downstream areas. Moreover, through the correlation distribution between the boreal winter NAOI and both contemporaneous and late North Atlantic SSTA corresponding to the selected NAO negative phase years (1979, 1981, 1984, 1985, 1986, 1995, 1997, 2000, 2002, 2003, 2005, 2009, 2010, and 2012), it can be found that the North Atlantic SSTA from boreal winter to late spring is manifested as a "-, +, -" triple-pole pattern characteristic (Fig. 4c to d), which further verifies that the early NAO negative signal can be stored in the North Atlantic SST and has an influence on the subsequent weather-climate over the surrounding and downstream regions.

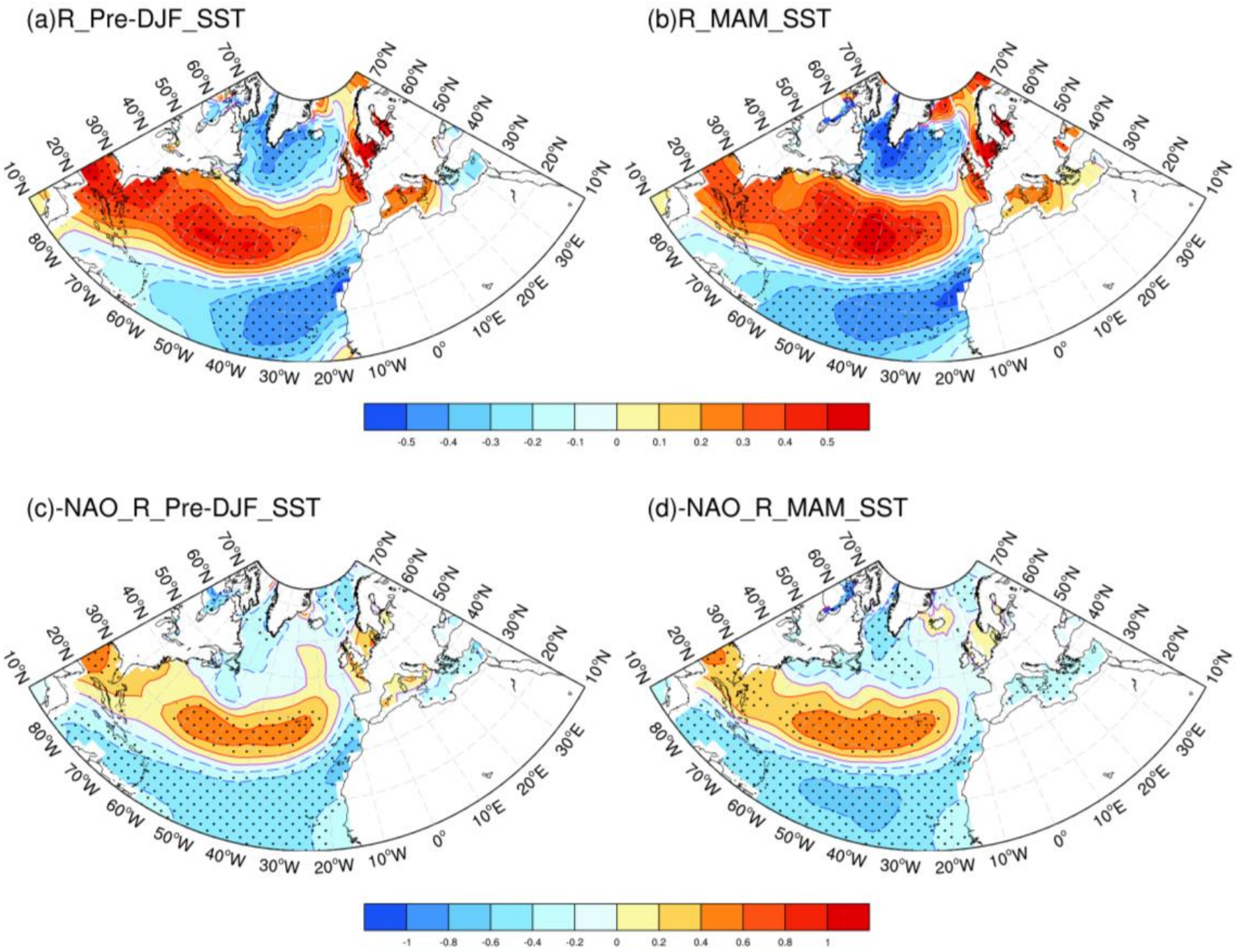

Figure 4. Correlations of the boreal winter NAOI from 1980-2020 with the SSTA during the previous period and the same period (a-b). (c-d) As in (a-b) but for the NAO negative phase years (1979, 1981, 1984, 1985, 1986, 1995, 1997, 2000, 2002, 2003, 2005, 2009, 2010, and 2012). Dotted blue, solid magenta and solid red lines indicate negative, zero and positive contour values, respectively. And the contour interval is 0.1 (a-b) and 0.2 (c-d), unit: °C. The black dotted areas are statistically significant at/above the 95% confidence level.

### 3.2 Atmospheric circulation evolution during DEs

In this paper, we selected 27 spring DEs in North China during the period of 1980-2020 based on the selection criteria, which are distributed over 17 years. It is found that there are shifts of the NAO from a positive to negative phase preceding all the DEs, highlighting the significant influence of the NAO negative phase on the occurrence of DEs. Among these, the 9 DEs (19800419, 19820408, 19820502, 19820508, 19850403, 19870317, 19980416, 20100320, 20130309) are striking based on the preliminary unusually strong winter NAO negative signal compared to that before the rest of DEs. At the climate-scale, the NAO intensity in boreal winter of the 9 DEs are all less than -0.5 standard deviations, which corresponds to the NAO negative phase years. Further analysis on the weather-scale indicates that the number of days with NAO negative phase (the value of NAOI less than -0.5) are above 35 days in the 3 months of boreal winter in the 9 DEs. Furthermore, the NAO negative phase is longer and more intense during these 9 DEs, while the NAO negative phase days in the remaining DEs are below 30 days (figure not shown). Therefore, to avoid information filtering caused by too many study cases and to reflect the statistical significance for studying the modulation effect of the NAO negative signal on the DEs in North China to the maximum extent, we chose these 9 DEs with the strongest influence of the NAO negative signal to study the evolution characteristics and synoptic causes of DEs in North China.

From the abnormal field of the spatial distribution of DA concentration synthesized by the selected DEs (Fig. 5), it can be found that before the outbreak of DEs, the positive anomalies of DA concentrations appear in the source regions (Xinjiang and Mongolia), and on “Day -1”, the positive anomalous field gradually expands and develops into a positive anomaly belt with maximum values of 180 mg $m^{-2}$. At this time, the DA concentrations on the north side of North China already show obvious positive anomalies, providing sufficient source contributions for the outbreak of DEs. On “Day 0”, the overall DA concentration anomaly value exceeds 180 mg $m^{-2}$ in North China, indicating that large-scale DEs have broken out. After the outbreak of DEs, the positive DA concentration anomalies in North China decrease rapidly, and the impact of DAs from Xinjiang and Mongolia on North China gradually weakens. Before the outbreak of DEs, the DA concentrations in Mongolia and Xinjiang, which are the primary source areas of DAs in East Asia (Zhang et al., 2003), gradually increase. In the growth process of DA concentrations, DAs are gradually

transported eastward to North China. Therefore, the outbreak of DEs in North China is mainly caused by the rapid increase in DA concentrations and their eastward transmission from Xinjiang and Mongolia.

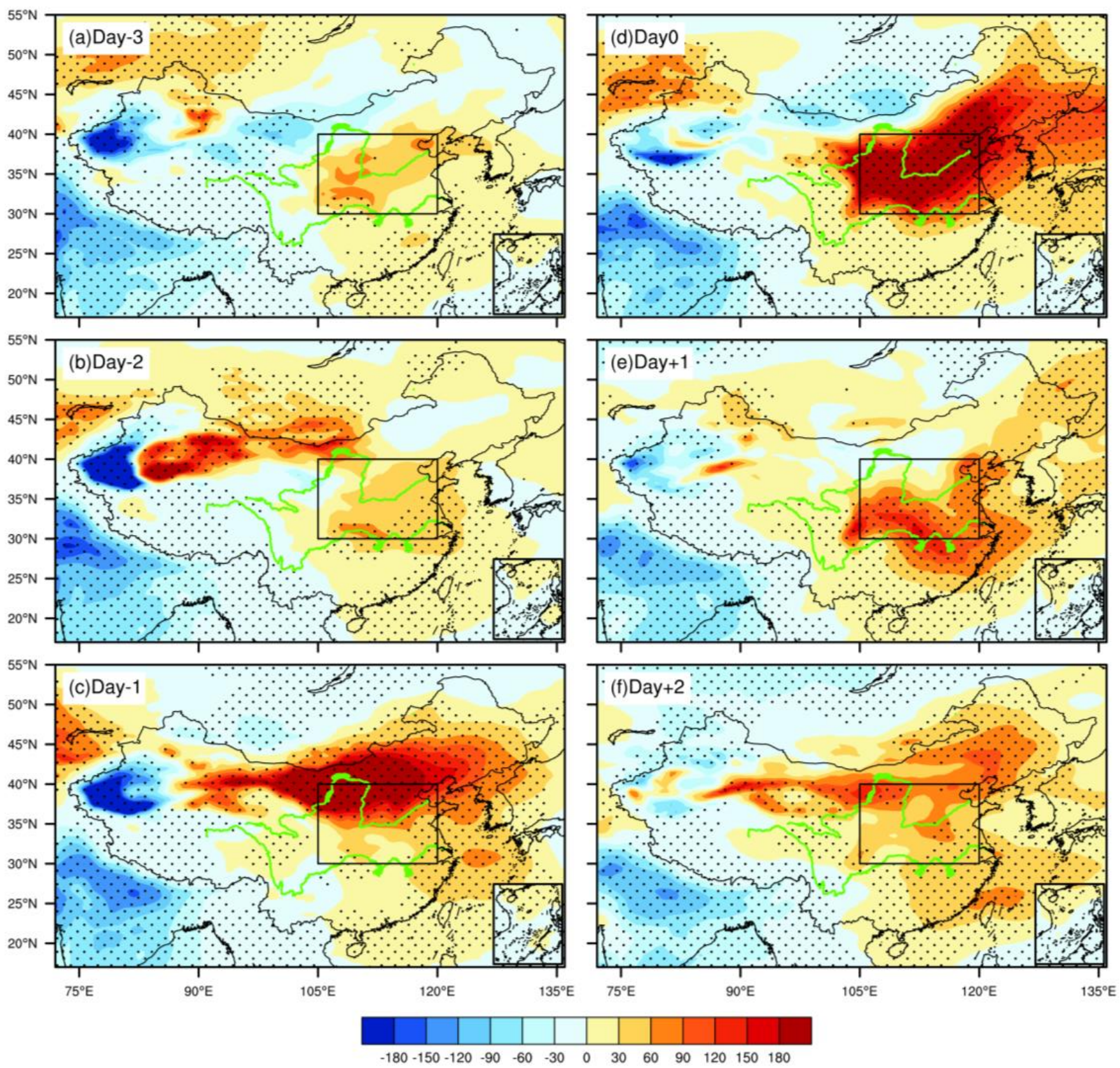

Figure 5. Spatial distribution of the dust column mass density anomalies in East Asia from "Day -3" to "Day +2" during DEs, unit: mg m$^{-2}$. The black box indicates the North China study area. The black dotted areas are statistically significant at/above the 95% confidence level.

The evolution of the 200 hPa atmospheric circulation during DEs is illustrated in Fig. 6, representing features of the wind field in high levels of the troposphere. On "Day -3", there are large-scale anomalous positive zonal winds in the Siberian area, with positive zonal wind anomalies of +9 m s$^{-1}$. From "Day -2" to "Day -1", the positive zonal wind anomalies move southward,

controlling the upstream area of North China (35-50°N, 70-110°E), as well as the desert source areas of Xinjiang and Mongolia. The zonal wind anomalies over the area of North China are in a "+, -, +" triple-pole pattern from the equator to 60°N, indicating that the boreal winter NAO negative signal can propagate to East Asia, resulting in changes in the East Asian subtropical jet stream (EASJS) and polar-front jet stream (PFJS) in late spring. In earlier investigations, similar findings were noted. For example, Shao and Zhang (2012) indicated that when the NAO is unusually strong (weak) in winter, the EASJS will strengthen (weaken) and the PFJS will weaken (strengthen). There are also remarkable features of the EASJS along 30°N, with two centres: the west centre is in northern Africa, with a central intensity of 40 m $s^{-1}$, and the east centre is situated in the western Pacific south of Japan, with a central intensity of 50 m $s^{-1}$. As a result of its far distance from the DA source regions in East Asia and North China, the western centre of the EASJS has a very limited impact on these regions. From "Day -3" to "Day -1", the eastern centre of the EASJS moves eastward, which leads to diminishing zonal winds at 200 hPa in North China, and the negative anomaly centre is lower than -9 m $s^{-1}$. From "Day 0" to "Day +2", the eastern centre of the EASJS stops moving eastward and recedes slightly westward, and the zonal winds over North China change to -6 m $s^{-1}$. Under the effect of vertical circulation, abnormal strong winds at high altitudes can influence middle and low altitude winds through downward momentum transmission, which may lead to the generation of windy weather near the surface (Wu et al., 2016).

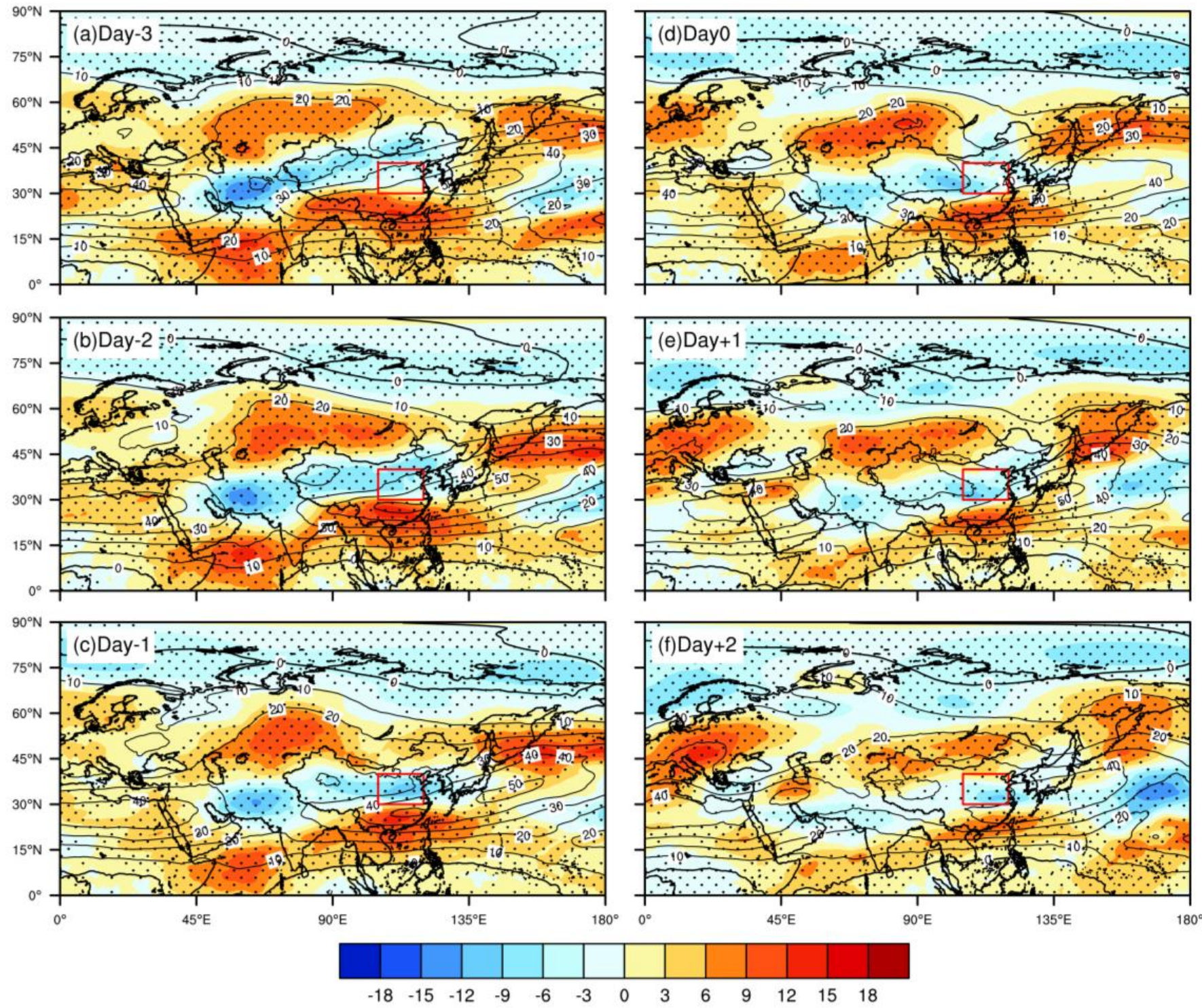

Figure 6. Spatial distribution of the 200 hPa winds (contour, unit: m s$^{-1}$) and their anomalies (shades, unit: m s$^{-1}$) from “Day -3” to “Day +2” during DEs. The red box indicates the North China study area. The black dotted areas are statistically significant at/above the 95% confidence level.

To further determine the impact of the upper-level wind speed on the middle-lower wind speed, the vertical pressure-latitude distribution of the mean zonal winds and the vertical winds are calculated over the DA source areas and North China (Fig. 7). As shown in Fig. 7a to f, before the outbreak of DEs, due to the downward momentum of the zonal winds at high altitudes above the DA source areas, the enhancement of the wind speed in the middle-low levels is evident (Wu et al., 2016), resulting in the generation of strong surface wind to meet the dynamic conditions for the uplift of local DAs. After the outbreak of DEs, the average zonal winds in the high levels and the downward momentum effect weaken in DA source areas. Compared with the mean zonal winds in the troposphere over North China (Fig. 7g to l), from “Day -3” to “Day -1”, corresponding to the weakening of the zonal winds in the high levels, the zonal wind speed in the middle-low levels

decreases due to momentum compensation for the zonal winds in the high levels (Li et al., 2015), which is conducive to the maintenance of DA concentration that is transported before the DE outbreak, as well as in preparation for the subsequent DE outbreak in North China. After the DE outbreak, the zonal winds at high altitudes on the north side of North China strengthen due to the increase in zonal winds in the middle-low levels, and the DA concentration begins to decrease under the effect of strong winds near the surface. It is also noted that during the whole evolution of DEs, the south side of North China is dominated by southerly winds, which have a certain blocking effect on the northward airflow carrying DAs and are favorable for maintaining the DA concentration in North China.

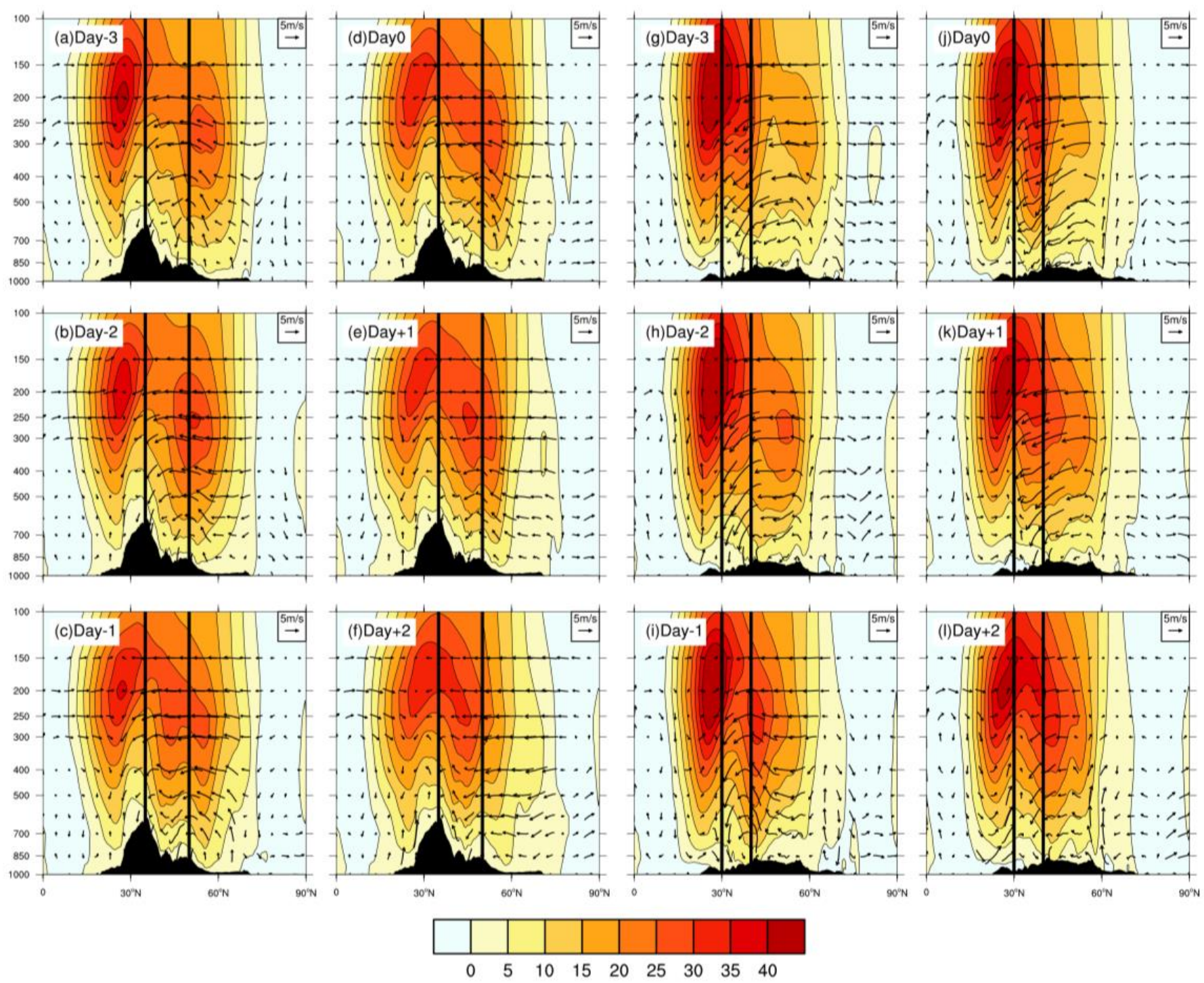

Figure 7. Pressure-latitude cross section of the mean zonal wind [$u$] (shaded, unit: m $s^{-1}$) and vertical wind (arrow, ($v$, $w$), $w$ expands 100 times, unit: m $s^{-1}$) averaged over 70°-110°E from “Day -3” to “Day +2” during DEs. The area surrounded by the black line is the range of the source regions of DA (a-f). (g-l) Same as in (a-f) but for 105°-120°E. The area surrounded by the black line is the North China study area.

In the evolution of atmospheric circulation of DEs at 500 hPa (Fig. 8a to c), the trough-ridge situation is characterized by two troughs and one ridge throughout the high latitudes of the Eurasian continent from “Day -3” to “Day -1”. The two troughs are located in the Black Sea (20-50°E, 30°-60°N) and the eastern part of Russia (105-130°E, 30°-60°N), and the ridge is situated in the Ural Mountains (60-80°E, 25-45°N). The western and eastern troughs both show negative variations before the outbreak of DEs, with maximum values of “-4 dagpm” and “-10 dagpm”, respectively, while the Ural ridge (UR) manifests as gradual enhancement with maximum values of “+6 dagpm”. From “Day 0” to “Day +2”, the intensities of the trough and ridge all gradually weaken (Fig. 8d to f). Before the outbreak of DEs, on account of the strengthening advancement of the trough-ridge situation, the middle-high latitudes are dominated by the strong meridional circulation, the northern cold air transport increases, and the southward invasion of the cold air enhances the local surface wind speed, leading to the uplift of DAs in the DA source areas. After the outbreak of DEs, due to the changes in the trough-ridge situation and the weakening of the UR, the cold air transport weakens, as well as the increasing dust activities and transportation. It should be noted that the UR during DEs is much less than a blocking high based on the intensity as well as the duration. The UR is established on “Day -1” and disappears basically on “Day +1”, which is also consistent with the feature that the maximum duration of DEs does not exceed 2 days (Li et al., 2022).

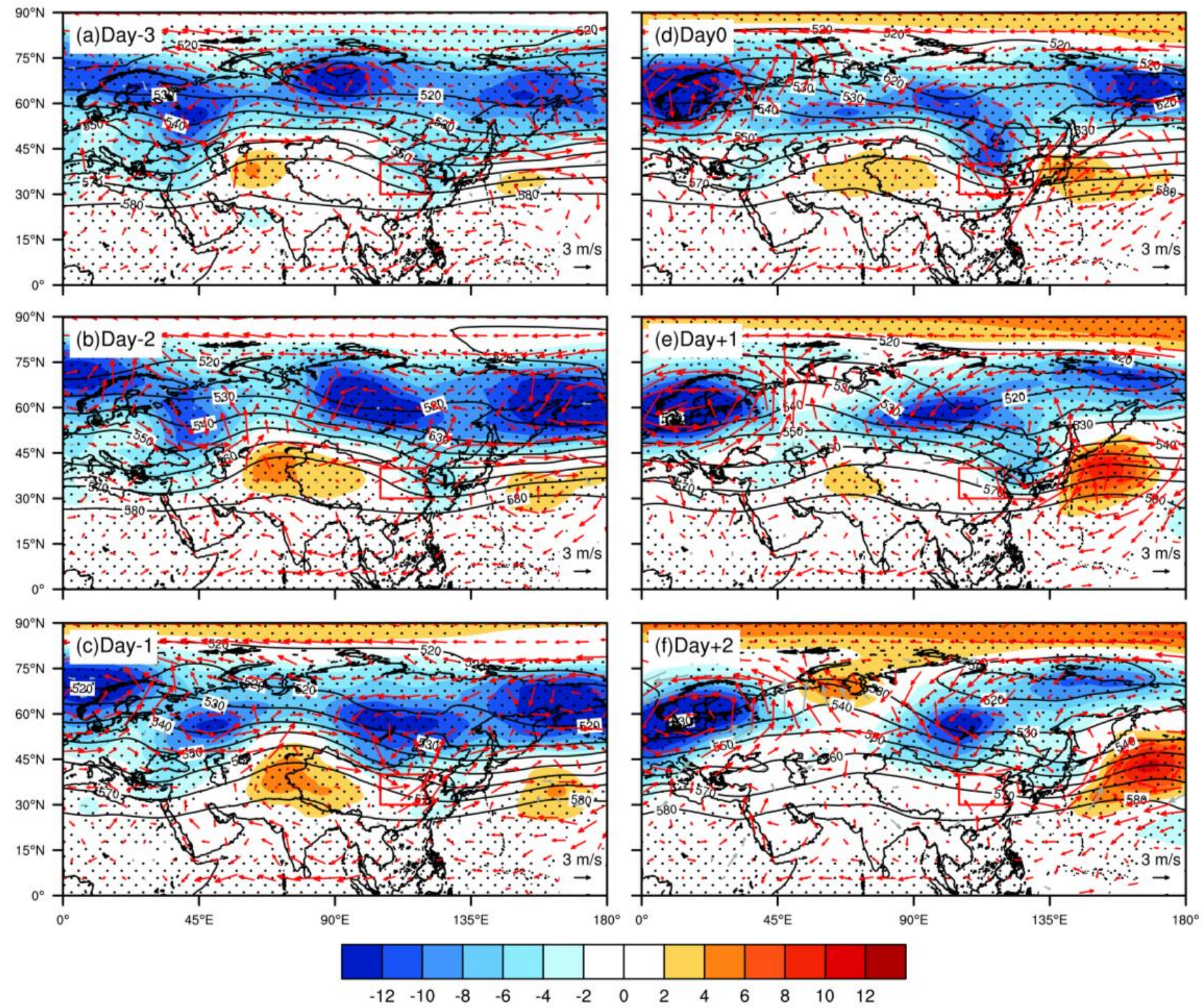

Figure 8. The spatial distribution of the 500 hPa geopotential height (contour, unit: dagpm) overlays its anomalies (shades, unit: dagpm) and the 850 hPa wind anomalies (arrows, unit: m $s^{-1}$) from "Day -3" to "Day +2" during DEs. The red box indicates the North China study area. The black dotted areas and red arrow areas are statistically significant at/above the 95% confidence level.

There is also a corresponding signal during DEs in the sea level pressure field (Fig. 9). On "Day -3", North China is controlled by an anticyclonic system dominated by sinking airflow. Therefore, poor precipitation conditions lead to the weak wet-deposition of DAs during transport (Kutiel and Furman, 2003). Meanwhile, the SH starts to appear in the upstream areas of North China. On "Day -2", the intensity of the SH is enhanced, with maximum values of "+6 hPa", and moves toward North China, while the intensity of the anticyclonic system controlling North China slightly weakens, and its position does not change significantly. In Mongolia, the MC starts to develop, accompanied by strong winds and vertical upward flows induced by the MC, which is conducive to the dynamic circumstances for the development uplift of DAs. On "Day -1", the location and

intensity of the SH both change little, while the intensity of the anticyclonic system controlling North China weakens and even tends to disappear, and the intensity of the MC increases and moves slightly southward. On "Day 0", the SH moves eastward to the territory of Mongolia, and the MC moves southeastward, which is conducive to the transport of DAs from Mongolia to North China. From "Day +1" to "Day +2", the intensities of both the SH and MC start to weaken, and the impact on North China begins to weaken. In summary, under the negative phase of the NAO, before the outbreak of DEs, due to the establishment, strengthening and southward movement of the SH and MC, there is a wide range of northerly winds and the outbreak of cold air to the south, which is advantageous for the uplift and transmission of DAs to North China. Simultaneously, North China is controlled by an anticyclonic system, leading to local weather, mainly sunny conditions and weak winds, which is also favorable for the transport of DAs to North China. After the outbreak of DEs, both the SH and MC start to weaken, indicating that the uplift of DAs in the DA source areas and the dust transmission activities to North China start to weaken.

From the above atmospheric circulation characteristics, under the NAO negative phase, the crucial synoptic systems leading to DE occurrences in North China are the abnormal winds by the anomalies of the upper-level EASJS, the UR, the SH and the frontal cyclone (MC) near the surface. In addition, transient eddy is crucial for the abnormal evolution of atmospheric circulation. The mechanisms of the formation of these synoptic system anomalies from the view of transient eddy are investigated to reveal the mechanism of the synoptic cause for the DEs in North China under the modulation of the NAO negative phase.

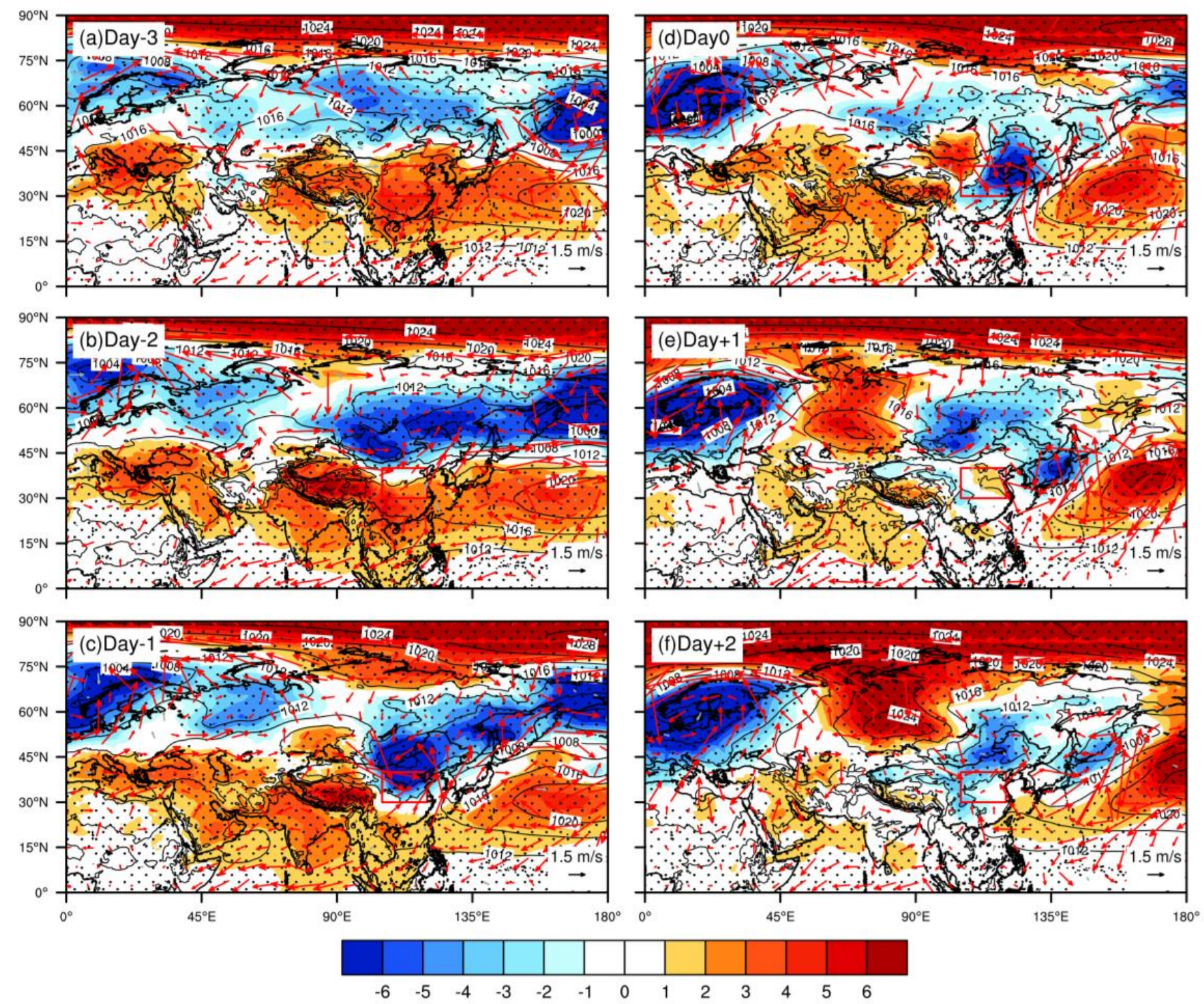

Figure 9. Spatial distribution of the sea level pressure (contour, unit: hPa) overlays with its anomalies (shades, unit: hPa) and the 1000 hPa wind anomalies (arrows, unit: m s$^{-1}$) from "Day -3" to "Day +2" during DEs. The red box indicates the North China study area. The black dotted areas and red arrow areas are statistically significant at/above the 95% confidence level.

### 3.3 Transient eddy flux transport characteristics during DEs

Transient eddy fluctuations contribute to the maintenance of the atmospheric energy balance through energy transport, and the energy transport process causes the divergence and convergence of energy and mass between different regions, thus forming new regions of forcing within the atmosphere (Li et al., 2019b). Transient eddy flux transport can enhance the positive pressure component of the mean airflow and compensate for ground friction, which has a significant effect on the maintenance of atmospheric circulation. For example, transient eddy momentum transport is associated with the development of blocking high generation, and momentum transport contributes differently during each period of its development (Li et al., 2019b). DE occurrences are inevitably

accompanied by tropospheric atmospheric circulation anomalies, and transient eddy flux transport plays a significant role in the process of atmospheric circulation. Therefore, the possible mechanisms of transient eddy momentum and heat transport during DEs under the NAO negative phase on the change in major synoptic system anomalies are further analysed and explored.

The "+" ("-") sign of $[u'v']$ implies the poleward (equatorward) transport of transient eddy momentum, and the "+" ("-") sign of $[u']$ represents the positive (negative) anomaly of the zonal winds. Analysing the transport features of transient eddy momentum within the active range of the UR during DEs (Fig. 10a to f), it is found that a pattern of "positive south and negative north" appears with poleward, equatorward momentum transport near 30°N and 40°N at approximately 200-500 hPa on "Day -3", respectively, while a $[u'v']$ convergence centre exists within the area of the UR. On "Day -2", the feature of $[u'v']$ in the Urals region changes from convergence to divergence, and on "Day-1", the "negative south and positive north" pattern of transient eddy momentum reaches the highest value during DEs, with maximum values of -24 $m^2\ s^{-2}$ and +30 $m^2\ s^{-2}$, respectively. After the outbreak of DEs, the intensity of the $[u'v']$ centre in the Urals region rapidly decreases, with no longer obvious $[u'v']$ transport. For the zonal wind anomalies $[u']$ (Fig. 10g to l), before the outbreak of DEs, $[u']$ in the Urals region there is a "negative south and positive north" mode, which is mainly controlled by negative $[u']$, with maximum values of -12 m $s^{-1}$, corresponding to divergence of $[u'v']$. From "Day 0" to "Day +1", the UR region gradually becomes dominated by the "+" signal of $[u']$. On "Day +2", as there is unobvious $[u'v']$ within the region of the UR, the intensity of $[u']$ weakens rapidly. Investigation of the transient eddy momentum transport characteristics and the changes in the zonal wind anomalies during DEs show that there is an obvious divergence centre of the transient eddy momentum within the range of UR before the outbreak of DEs, corresponding to zonal wind weakening and the establishment of meridional circulation. Therefore, transient eddy momentum transport has an indirect influence on the establishment and advancement of the UR (Li et al., 2022). To obtain a better understanding of the role of transient eddy momentum on anomalies of the wind field during DEs, the $[u'v']$ characteristics of the DA source areas are further analysed (Fig. 11a to f). The transient eddy momentum pattern of "positive south and negative north" in the DA source areas is gradually displayed at 200-500 hPa from "Day -3" to "Day -1", with maximum values of +36 $m^2\ s^{-2}$ and -36

m² s⁻², respectively. After the outbreak of DEs, the source areas of DAs are mainly controlled by the weak $[u'v']$ negative centre, with unobvious transient eddy momentum. For the zonal wind anomalies $[u']$ (Fig. 11g to l), before the outbreak of DEs, the DA source regions gradually become dominated by positive $[u']$, with maximum values of +10 m s$^{-1}$ on account of the convergence of transient eddy momentum, facilitating the uplift of DAs. After the outbreak of DEs, $[u']$ still predominantly increases, indicating that there are more complex reasons for the zonal wind changes than the transport of transient eddy momentum, which deserves in-depth analysis later.

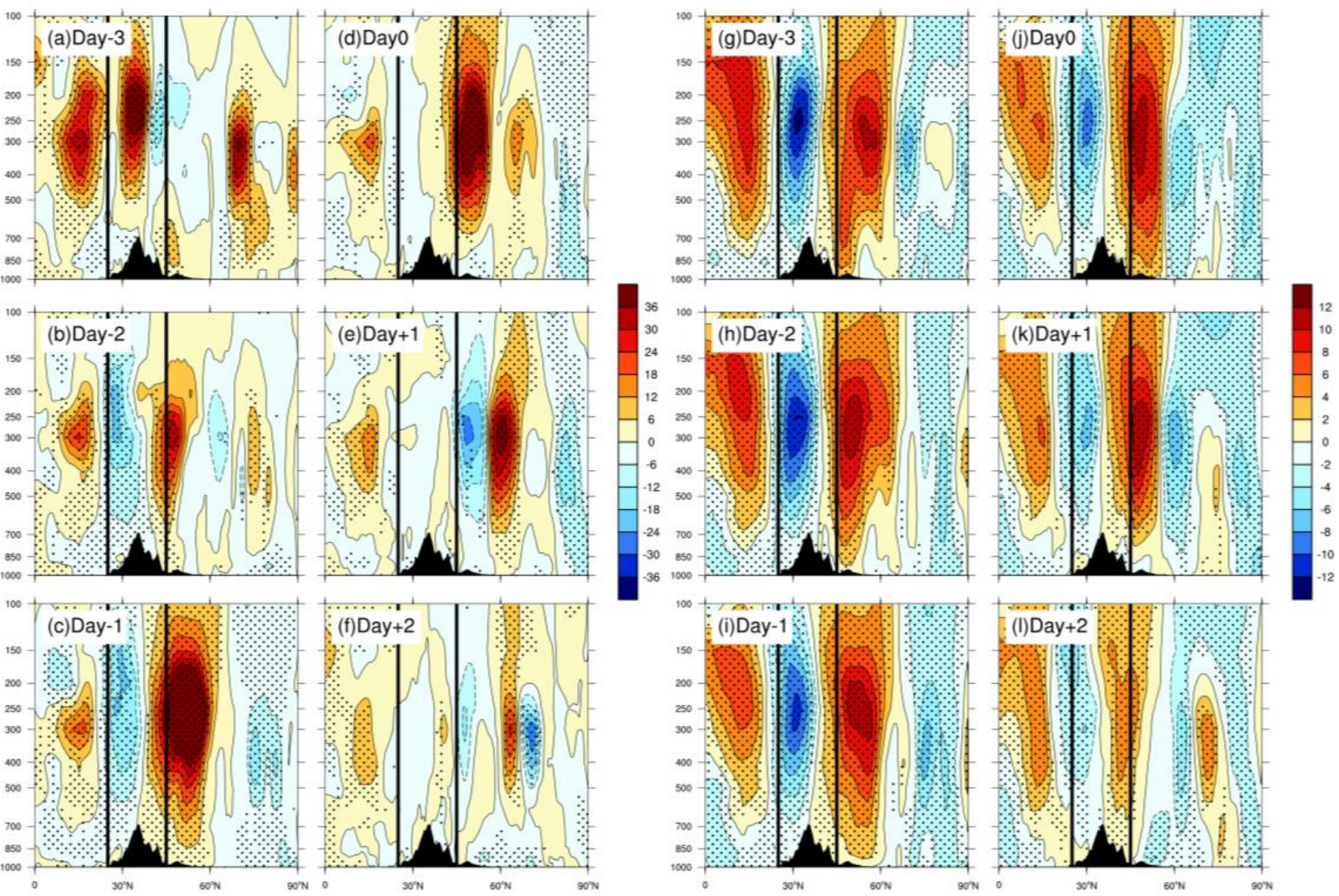

Figure 10. Pressure-latitude cross section of the mean transient momentum $[u'v']$ (unit: m$^2$ s$^{-2}$) (a-f) and the zonal wind anomalies $[u']$ (unit: m s$^{-1}$) (g-l) averaged over 60°-80°E from "Day -3" to "Day +2" during DEs. The area surrounded by the black line is the range of the UR. The black dotted areas are statistically significant at/above the 95% confidence level.

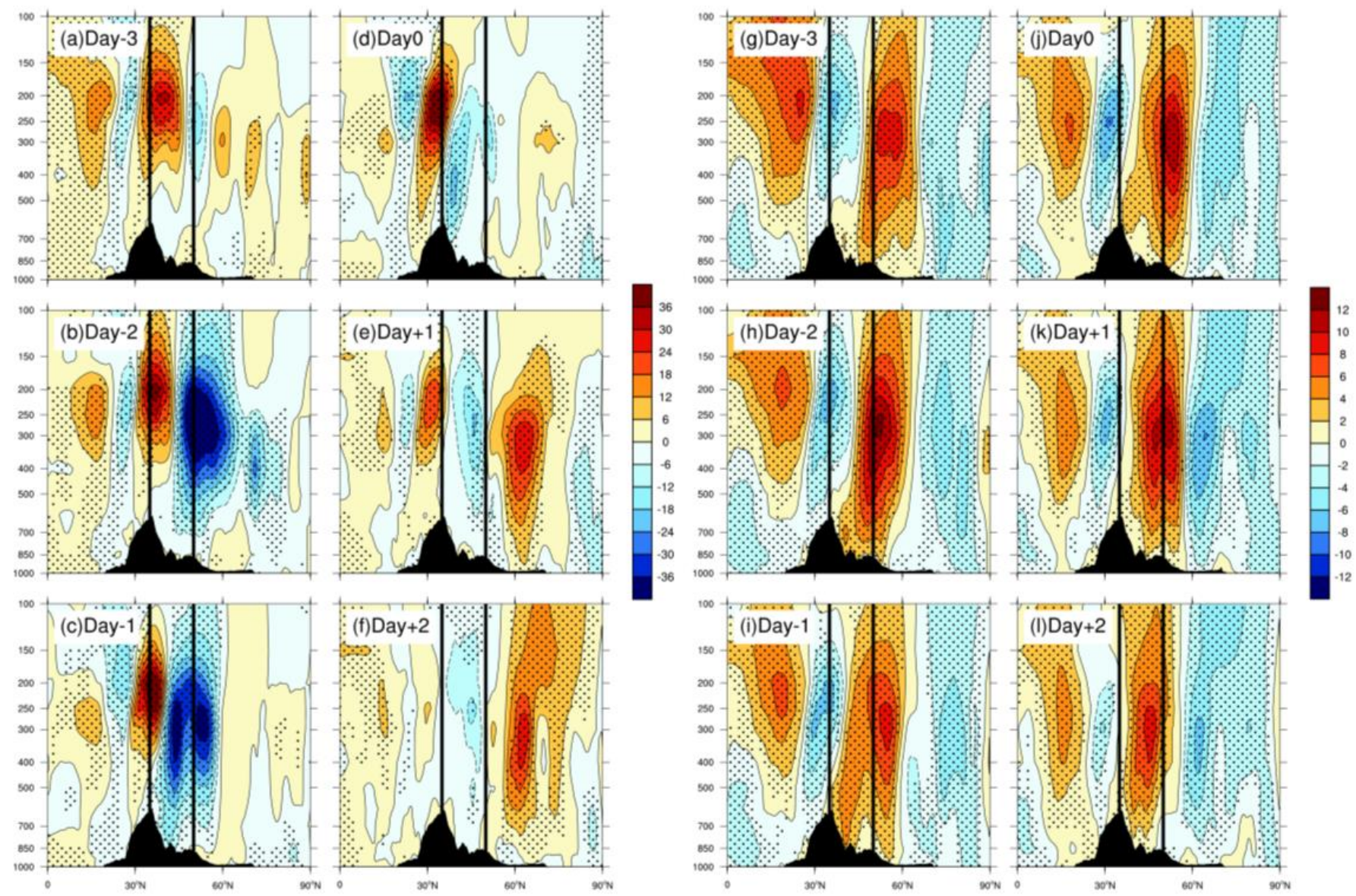

Figure 11. Pressure-latitude cross section of the mean transient momentum $[u'v']$ (unit: $m^2\ s^{-2}$) (a-f) and the zonal wind anomalies $[u']$ (unit: $m\ s^{-1}$) (g-l) averaged over 70°-110°E from "Day -3" to "Day +2" during DEs. The area surrounded by the black line is the range of the source regions of DA. The black dotted areas are statistically significant at/above the 95% confidence level.

Atmospheric circulation also heavily depends on transient eddy heat transport (Li et al., 2019b; Li et al., 2022). Similar to the features of transient eddy momentum transport, the "+" ("-") sign of $[v'T']$ describes the poleward (equatorward) transport of transient eddy heat, and the "+" ("-") sign of $[T']$ implies an increase (decrease) in temperature. In the analysis of the $[v'T']$ characteristics within the range of UR during DEs (Fig. 12a to f), it is found that the convergence of transient eddy heat is dominant at 200-500 hPa on "Day -3", with maximum values of +12 $m\ s^{-1}$ K and -6 $m\ s^{-1}$ K, respectively. During the subsequent "Day -2" to "Day -1", the UR area is dominated by a strong positive $[v'T']$ centre, which also exhibits the convergence features of transient eddy heat. There are unobvious features of $[v'T']$ in the developmental range of the UR from "Day 0" to "Day +2". In the analysis of the temperature anomalies $[T']$ (Fig. 12g to l), it is shown that before the outbreak of DEs, the area with developed UR is in the convergence region of transient eddy heat, and $[T']$ becomes more visible when the transient eddy heat transport increases, showing a "negative south

and positive north" pattern below 300 hPa. The temperature gradient ($-\partial T / \partial y$) is weakened here by the thermal wind formula ($u_T = -R \cdot \partial T / \partial y$, where $u_T$, $R$, and $T$ demonstrate the thermal wind, gas constant and temperature, respectively), and the thermal wind will be weakened due to the decline in the temperature gradient. When the thermal wind is weakened, the zonal winds are also weakened, which facilitates the establishment of meridional circulation and the development of a blocking situation (Li et al., 2019b). After the outbreak of DEs, $[T']$ decreases due to the weakening of the transient eddy heat transport. From "Day 0" to "Day +1", the negative $[T']$ anomaly centre appears north of the UR, and the negative $[T']$ centre gradually moves southward, which corresponds to the increase in the temperature gradient here and is not favorable to the development of the UR. The transient eddy heat transport also has an impact on the wind field for the source regions of DAs (Fig. 13a to c). Before the outbreak of DEs, the $[v'T']$ over the DA source areas gradually evolves to be dominated by the convergence above 300 hPa, while the divergence of $[v'T']$ is dominated below 300 hPa. The strongest $[v'T']$ convergence and divergence characteristics are both reached on "Day -1". After the outbreak of DEs (Fig. 13d to f), the DA source regions are controlled by a positive $[v'T']$ centre, with the convergence of $[v'T']$ on "Day 0". From "Day +1" to "Day +2", there are unobvious $[v'T']$ features over the source areas of DAs. Analysis of the temperature anomalies $[T']$ (Fig. 13g to l) reveals that below 300 hPa in the DA source regions before the outbreak of DEs, the transient eddy heat transport weakens with time due to the "positive south and negative north" mode of $[T']$, with maximum values of +2 K and -2 K, respectively. The temperature gradient is enhanced, and the development of thermal wind leads to the enhancement of zonal winds. In addition, the zonal winds in the middle-low levels are enhanced by the enhanced thermal wind, and the momentum downward transport of zonal winds in the high levels further enhances the zonal winds in the middle-low levels, which is conducive to the uplift of DAs. In addition, the convergence of $[v'T']$ is unfavorable to the development of the zonal winds above 300 hPa according to the thermal wind formula, which hinders the momentum downward transport of the zonal winds in the high levels in the DA source areas. Transient momentum transport has a greater effect on the variability of zonal winds at high levels than transient eddy heat transport (Solomon, 1997).

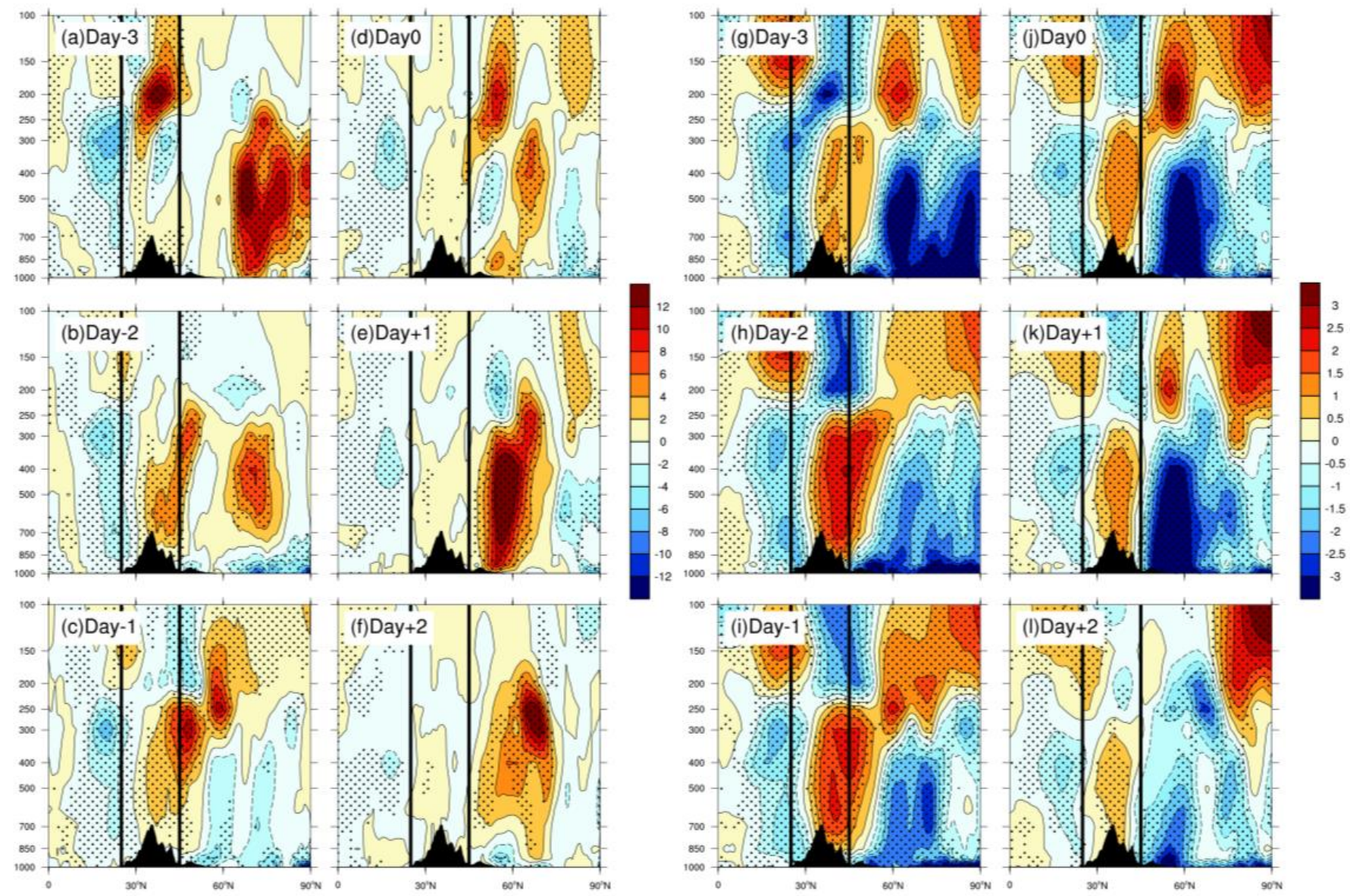

Figure 12. Pressure-latitude cross section of the mean transient heat $[v'T']$ (unit: m s$^{-1}$ K) (a-f) and temperature anomalies $[T']$ (unit: K) (g-l) averaged over 60°-80°E from "Day -3" to "Day +2" during DEs. The area surrounded by the black line is the range of the UR. The black dotted areas are statistically significant at/above the 95% confidence level.

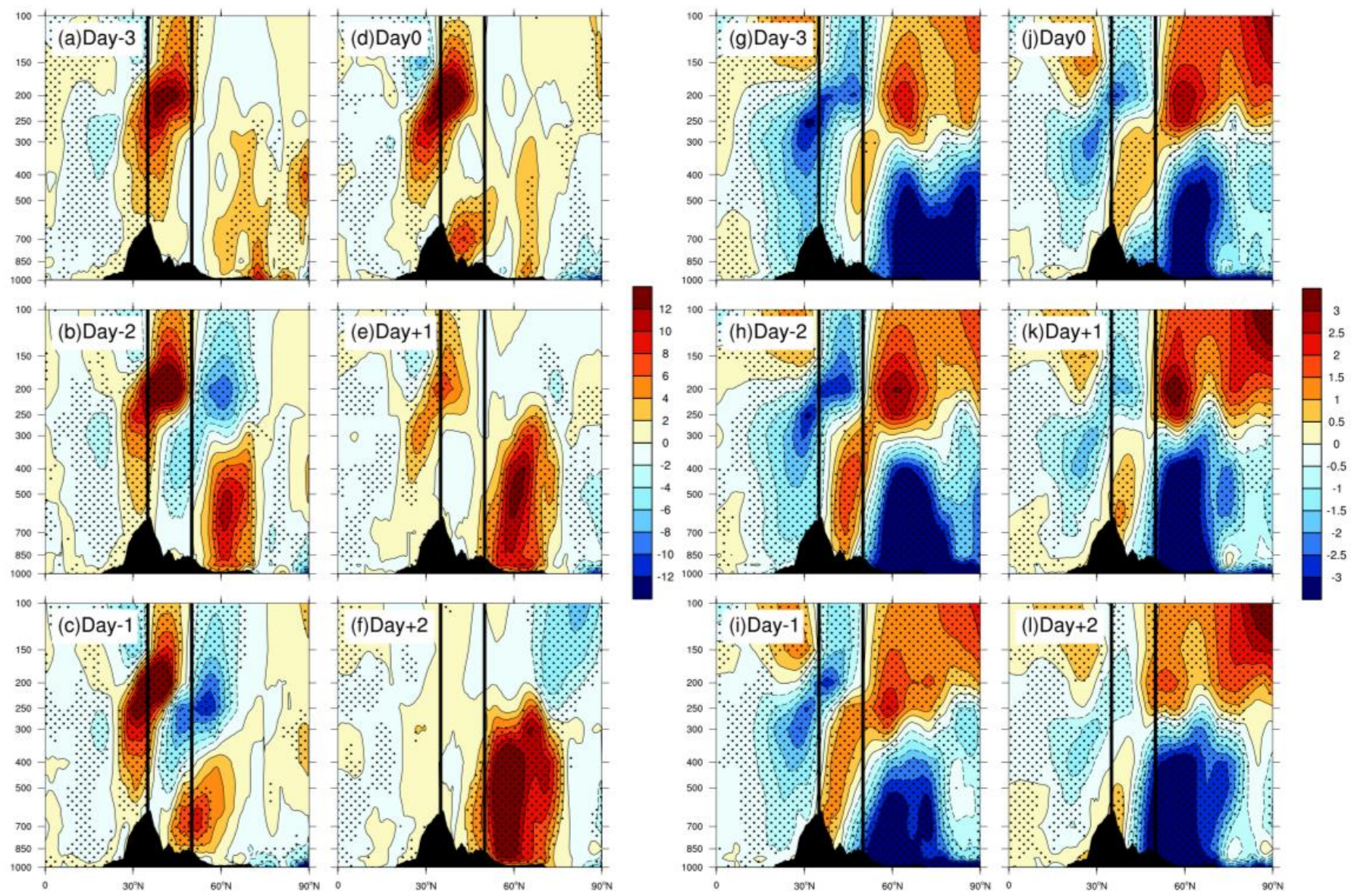

Figure 13. Pressure-latitude cross section of the mean transient heat $[v'T']$ (unit: m s$^{-1}$ K) (a-f) and temperature anomalies $[T']$ (unit: K) (g-l) averaged over 70°-110°E from "Day -3" to "Day +2" during DEs. The area surrounded by the black line is the range of the source regions of DA. The black dotted areas are statistically significant at/above the 95% confidence level.

The transient eddy flux transport characteristics during DEs show that the changes in $[u'v']$ and $[v'T']$ in the DA source regions and the active range of the UR cause both energy and mass divergence in North China, leading to diminishing zonal winds at high levels over North China during DEs. Meanwhile, the zonal winds in the middle-low levels weaken through vertical circulation, accompanied by southerly airflow on the south side of North China, facilitating DA concentration.

Results of the previous analysis of the transient eddy flux transport characteristics indicate that the evolution of the UR may be explained by the divergence of transient eddy momentum and the convergence of transient eddy heat by the thermal wind formula. By using the equation for the quasi-geostrophic potential, this mechanism can also be verified (Gary, 2011) as follows:

$$\left[\nabla^2 + \frac{\partial}{\partial p}\cdot\left(\frac{f^2}{\sigma}\cdot\frac{\partial}{\partial p}\right)\right]\cdot\frac{\partial \Phi}{\partial t} = -f\cdot\vec{V_g}\cdot\nabla\left(\xi_g + f\right) - \frac{\partial}{\partial p}\cdot\left[-\frac{f^2}{\sigma}\cdot\vec{V_g}\cdot\nabla\left(-\frac{\partial \Phi}{\partial p}\right)\right] \quad (3)$$

where $f$, $\Phi$, and $\vec{V_g}$ denote the Coriolis parameter, geopotential height, and geopotential wind, respectively, and $\sigma = -\frac{RT}{p}\cdot\frac{\partial \ln\theta}{\partial p}$, $\xi_g = \frac{\partial v_g}{\partial x} - \frac{\partial u_g}{\partial y} = \frac{1}{f}\cdot\nabla^2\Phi$ denote the static stability and the geostrophic relative vorticity, respectively. The first and second terms on the right-hand side of the equal sign of Eq. (3) have the following equivalence after simplification:

$$-\frac{\partial\Phi}{\partial t} \propto -\vec{V_g}\cdot\nabla\xi_g \tag{4}$$

$$-\frac{\partial\Phi}{\partial t} \propto \frac{\partial}{\partial p}\left(-\vec{V_g}\cdot\nabla T\right) \tag{5}$$

On the trough-ridge lines, since the vorticity advection in Eq. (4) is zero and the height of the isobaric surface has no change, it only moves the ridges and does not change the strength of the ridges. The temperature advection in Eq. (5) mainly acts on the trough-ridge lines, which affects the development of the trough-ridge intensity but has no influence on its movement. In this study, we mainly consider the change in trough-ridge intensity during DEs by noting the effect of temperature advection. Temperature advection generally decreases with height, so it is sufficient to discuss only the signs of positive and negative temperature advection here. Analysis of transient eddy heat transport characteristics reveal that the transport of transient eddy heat in the UR region before the outbreak of DEs will lead to uneven temperature distribution, which in turn leads to the development of warm advection. The analysis of the quasi-geostrophic potential theory shows that the development of warm advection in this region is beneficial to the development of the UR and to the establishment of a temperature ridge. After the outbreak of DEs, the temperature advection starts to weaken as [$v'T'$] no longer change within the area of UR, and thus, the intensity of UR also starts to weaken.

Meanwhile, the active development of temperature advection and vorticity advection due to transient eddy heat transport and the enhancement of UR, based on the $\Omega$ equation in the quasi-geostrophic theory (Gary, 2011), can also explain abnormal evolutions of the SH as well as the MC near the surface during DEs, as follows:

$$\left(\nabla^2 + \frac{f^2}{\sigma}\cdot\frac{\partial^2}{\partial p^2}\right)\omega = \frac{f}{\sigma}\cdot\frac{\partial}{\partial p}\left[\vec{V_g}\cdot\nabla\left(\xi_g + f\right)\right] + \frac{1}{\sigma}\cdot\nabla^2\left[\vec{V_g}\cdot\nabla\left(-\frac{\partial\Phi}{\partial p}\right)\right] \tag{6}$$

where $f$, $\omega$, $\vec{V_g}$, and $\Phi$ denote the Coriolis parameter, vertical velocity, geopotential height,

and geopotential wind, respectively, and $\sigma = -\frac{RT}{p} \cdot \frac{\partial \ln \theta}{\partial p}$, $\omega = \frac{dp}{dt}$, and $\xi_g = \frac{\partial v_g}{\partial x} - \frac{\partial u_g}{\partial y} = \frac{1}{f} \cdot \nabla^2 \Phi$ denote the static stability, vertical velocity, and geostrophic relative vorticity, respectively. After simplification, the first and second terms in Eq. (6) to the right of the equal sign are equivalent to the following:

$$\omega \propto \frac{\partial}{\partial p} \cdot \left[ -\overrightarrow{V_g} \cdot \nabla \xi_g \right] \quad (7)$$

$$\omega \propto \overrightarrow{V_g} \cdot \nabla T \quad (8)$$

According to Eq. (7) and (8), the relationship between the vertical velocity and vorticity advection and temperature advection can be obtained, respectively. When the positive (negative) vorticity advection increases with height, the degree of counterclockwise (clockwise) rotation increases there, and the divergence (convergence) increases with height under the Coriolis force, which will produce the rising (sinking) motion. As the vorticity advection generally increases with height, it is sufficient to discuss only the signs of positive and negative vorticity advection here. In short, it is expressed as warm (cold) advection corresponding to the rising (sinking) motion. Before the outbreak of DEs, the northwest winds in front of the UR are strengthened along with the enhancement of the ridge, and the northwest winds correspond to the development of negative vorticity advection and cold advection. According to the quasi-geostrophic theory, a sinking motion will be generated in front of the UR, which corresponds to pressurization at the surface and will promote the establishment, strengthening, and southward movement of the SH. Similarly, the Mongolian region is located in front of the trough at this time, controlled by the southwesterly winds. Corresponding to the development of positive vorticity advection and warm advection here, an upward motion will be generated in Mongolia, corresponding to depressurization at the surface, which will promote the establishment, enhancement, and southward movement of MC there. After the outbreak of DEs, both the SH and MC start to weaken by reversal impacts of the temperature and vorticity advections due to the weakening of the trough-ridge situation at a deeper level due to the change in transient eddy transport conditions.

## 4 Conclusions and discussions

As a significant extra-equatorial mode of low-frequency atmospheric variability with a periodic

signal on the seasonal-scale in the NH and a dominant mode of seasonal to annual winter semi-annual variability, the NAO has an influence on the DEs in East Asia, with significant regional characteristics. The spring DA concentration in North China, a non-dust source region, exhibits high values and strong annual variability. In this study, it is found that the boreal winter NAO negative signal has a significant effect on the DEs in late spring in North China. Modulated by the NAO negative signal, the tropospheric weather situation shows obvious anomalies, and the evolution mechanism can be revealed from the perspective of transient eddy transport.

The boreal winter NAO negative signal through the "capacitor effect" of the North Atlantic prolongs its atmospheric signal and influences the late spring DEs in North China. Under the modulation impact of the upstream NAO negative signal, the zonal winds at 200 hPa above the DA source regions gradually increase before the outbreak of DEs, and the increased zonal winds in the high levels transport the momentum to the zonal winds in the middle-low levels through the vertical circulation, which is favorable to the uplift of DAs in the DA source areas. Meanwhile, the zonal wind in the high levels is dominated by negative anomalies over North China, which has a counter effect on the zonal winds in the middle-low levels by vertical circulation and facilitates the maintenance of DA concentrations in North China. At 500 hPa, meridional circulation is obvious in Eurasia continent, and the UR is established, which is favorable for the accumulation of upper-level cold air and the activities of lower-level cold air. Near the surface, the SH and MC are established, enhanced and move southward, providing the northerly airflow in front of the DA source regions, guiding the cold air at high latitudes southward and acting favourably to the uplift and transmission of DAs to downstream North China. Simultaneously, the south side of North China is dominated by southerly winds, which have a blocking effect on the northerly airflow carrying DAs, facilitating the maintenance of DA concentrations in North China. After the outbreak of DEs, both favorable atmospheric circulations weaken gradually, and the DAs also start to decrease in North China.

Transient eddy has an important effect on the synoptic evolution of DEs through the modulation of the NAO negative signal. In advance of the outbreak of DEs, the transient eddy momentum at 300-500 hPa above the DA source region is dominated by convergence, and thus, the zonal winds are enhanced. Meanwhile, the characteristics of transient eddy heat over the source areas of DAs are mainly divergence, and the temperature anomalies are in a "positive south and

negative north" pattern, resulting in enhancement of the temperature gradient and the zonal winds. Therefore, the zonal winds over the DA source regions increased by both the downward momentum and thermal wind, which is favorable to the uplift of DAs. Within the area of UR, the transient eddy momentum (heat) diverges (converges). The divergence of transient eddy momentum will lead to the weakening of zonal winds, while the decrease in zonal winds and the reduction in temperature gradients are both favoured by the convergence of transient eddy heat. The establishment and growth of the meridional circulation are significantly influenced by both, which is conducive to the development of the UR. The changes in upstream transient eddy flux transport cause both energy and mass divergence in North China, resulting in diminishing zonal winds during DEs. After the outbreak of DEs, the transient eddy flux transport characteristics in both the Ural region and the source areas of DAs gradually weaken, and the effect on the zonal winds within these regions is also reduced.

In this study, we examine how the late spring DEs in North China are impacted by the boreal winter NAO negative signal and the corresponding synoptic mechanism from the view of transient eddy flux transport on the weather-scale by selecting 9 DEs from 1980 to 2020. Our focus is different from previous analyses of the NAO and the DEs in China on the seasonal climate-scale (Tang et al., 2005; Zhao et al., 2012). The findings demonstrate that the boreal winter NAO negative signal can store its signal in the North Atlantic as a triple-pole structure of "-, +, -", which then releases in late spring and affects the anomaly of atmospheric circulation in the troposphere by transient eddy flux transport, as well as the zonal winds over the DA source regions and North China. The thermal wind principle and the quasi-geostrophic theory can both explain this mechanism. According to changes in the temperature gradient, the thermal wind principle directly explains variations in the wind fields by the convergence and divergence of the thermal and momentum transient eddy flux during DEs. The quasi-geostrophic theory can illustrate the abnormal formation mechanism of synoptic systems in the whole troposphere during DEs. To be more precise, the influence of the NAO negative signal in transient eddy fluxes causes temperature and vorticity advection to develop; therefore, both mid-upper-level systems (the UR) and surface systems (the SH and MC) strengthen, which can be explained by the height tendency equation and $\Omega$ equation, respectively. The development of the above synoptic systems is favorable to the uplift of DAs in the DA source areas and the transmission process to North China. Meanwhile, the southward

airflow on the south side of North China is favorable for maintaining the stable high value of DA concentration for 1-2 days. The results are illustrated in Fig. 14, which displays the main synoptic systems and transient eddy flux transport characteristics during DEs against the background of the NAO negative phase. The result is helpful for a thorough comprehension of the mechanism underlying the formation of DEs in eastern China and could serve as a point of reference for the seasonal forecasting of DEs (Shao et al., 2003; Hong et al., 2019).

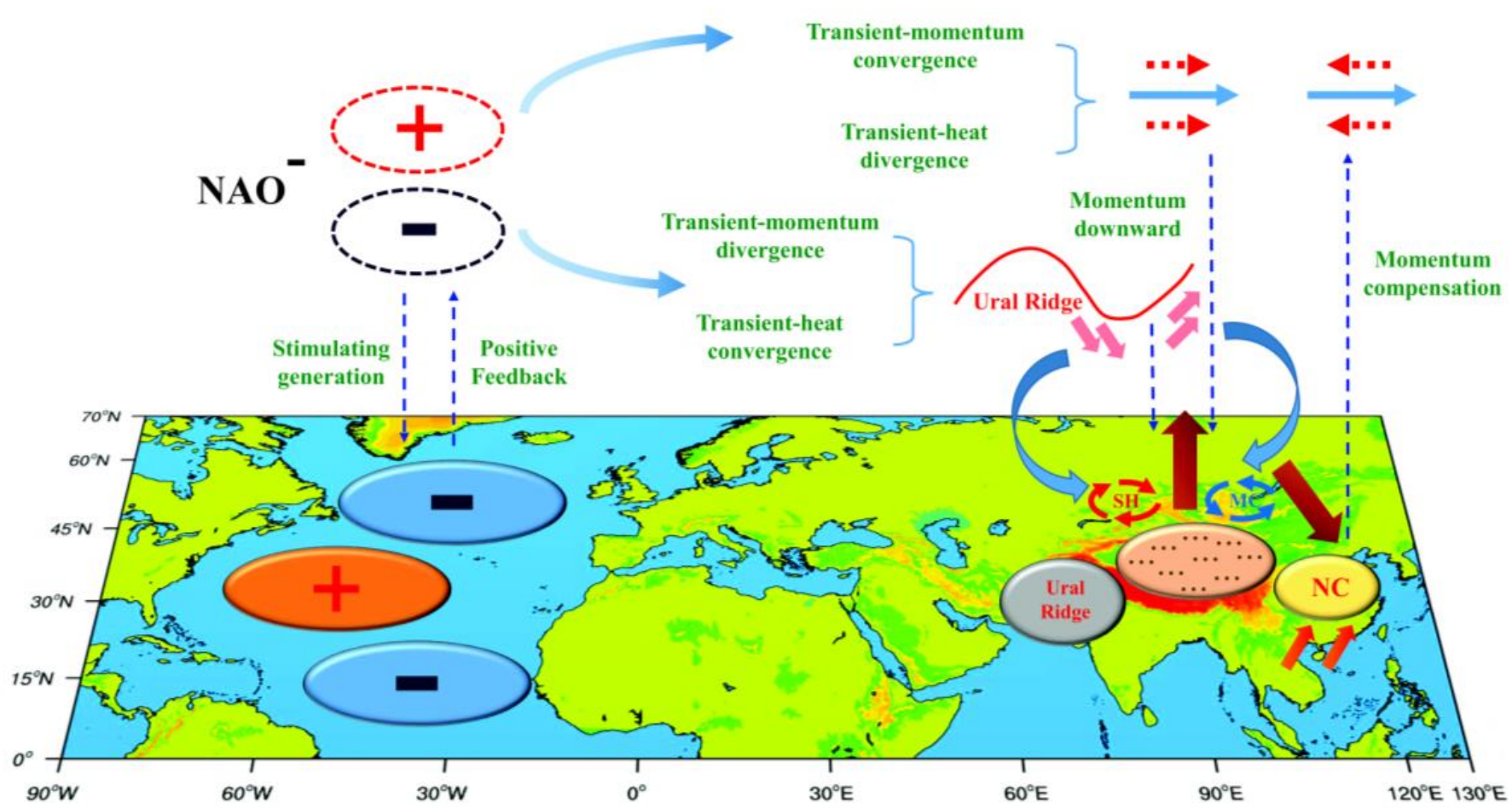

Figure 14. A schematic diagram of the main synoptic system anomalies and the role of transient eddy fluxes during DEs.

In addition to transient eddy, stationary eddy is also crucial in the development of atmospheric circulation, and the energy transport of stationary eddy differs from that of transient eddy, and has been applied to the analysis of atmospheric circulation processes such as the UR (Li et al., 2019c). To further advance our knowledge of dust hazards at the regional-scale in China, the impacts of various forms of energy transport on the synoptic systems of DEs in China will be concurrently considered in future work. Globally, the majority of DAs are found in arid and semi-arid areas, and the most widespread and longest-lasting source regions of DAs are located in the NH, forming a dust belt starting from the west coast of northern Africa and extending through the Arabian Peninsula and central and southern Asia to eastern China (Washington et al., 2003; Ginoux et al., 2012). In this study, the contribution of DA source regions in East Asia to the DEs in North China

is mainly considered, but insufficient consideration is given to other DA source regions. In further studies, more study methods, such as numerical simulations, should be applied to fully explore the role of DA transport from the NH source areas of DAs to China and to elucidate the details of the development of DEs in China. In addition, the DEs in China are also impacted by other factors that are related to the NAO, such as the ENSO (Li et al., 2021b; Yang et al., 2022). Previous research has demonstrated that central Pacific El Niño events can stimulate the negative phase of the NAO by transmitting the Pacific signal to the north Atlantic through a "subtropical bridge" mechanism, while this association is insignificant for eastern Pacific El Niño events (Graf and Zanchettin, 2012). Zhang et al. (2015) also discovered that the north Atlantic jet stream strengthens and the NAO exhibits a positive phase during central Pacific La Niña events, whereas the north Atlantic jet stream weakens and the NAO exhibits a negative phase during eastern Pacific La Niña events. In the context of global warming, the SSTA of the tropical Pacific mainly exhibits a cold tongue mode, and the positive phase of the cold tongue mode can easily stimulate central Pacific El Niño events (Li et al., 2017). However, it is not clear how the connection between the NAO and ENSO will evolve under global warming. Therefore, it is worthwhile to continue researching the synergistic effect of the NAO and ENSO on the DEs in China.

**Code and data availability.** The MERRA-2 dust aerosol concentrations dataset can be downloaded from https://disc.gsfc.nasa.gov/datasets?project=MERRA-2 (last access: 12 January 2023). The atmospheric reanalysis datasets, including the wind field, geopotential height field, sea level pressure field, temperature field, and vertical velocity field can be downloaded from https://cds.climate.copernicus.eu/#!/search?text=ERA5&type=dataset (last access: 12 January 2023). Our results can be made available upon request.

**Author contributions.** YL, FLX, and JF conceptualized and designed the research. FLX synthesized and analyzed the data. YL, FLX, and JF produced the figures. CL and WJZ contributed to the MERRA-2 dataset retrieval. All the authors including MYD, WJS discussed the results and wrote the paper.

**Competing interests.** The authors declare that they have no conflict of interest.

**Disclaimer.** Publisher's note: Copernicus Publications remains neutral with regard to jurisdictional claims in published maps and institutional affiliations.

**Acknowledgements.** This research has been supported by the National Key Research & Development (R&D) Program of China (Grant/Award Number: 2019YFA0606801).

**Financial support.** This research has been supported by the National Key Research & Development (R&D) Program of China (Grant/Award Number: 2019YFA0606801).

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
