# Peer review of "on the spring dust aerosols over North China"

_EGUsphere, 2023_

## Author Comment (AC1)

**Response to Comments of Reviewer 1**

**Manuscript number**: egusphere-2023-52

**Author(s)**: Yan Li, Falei Xu, Juan Feng, Mengying Du, Wenjun Song, Chao Li, and Wenjing Zhao

**Title**: Influence of the previous North Atlantic Oscillation (NAO) on the spring dust aerosols over North China

**General comments:**

Using the station observations and multi reanalysis data, the authors investigated the possible influences of the previous NAO on the dust aerosols over northern China. Authors concluded that the boreal winter NAO has significant impacts on the following spring dust aerosols, particularly during the negative phase of the NAO. The thermal and dynamical processes relevant to the anomalous NAO on the circulation are analyzed, and it indicated that the impact of transient eddy fluxes transport plays important role in the formation of the dust aerosol events. I agree with the authors that the influence of previous NAO on dust aerosols in China cannot be ignored. And it is an important interdisciplinary issue that needs more attention and deep research. Overall the manuscript is well written and clear, and the figures are also appropriate and clear. However, there are some problems with this manuscript, and it cannot be accepted by ACP as it is now. Some specific comments or suggestions are listed as follows.

**Response:**

Thanks to the reviewer for the helpful comments and suggestions. We have revised the manuscript seriously and carefully according to the reviewer's comments and suggestions. The point-to-point responses to the comments are listed as follows.

**Specific comments are as follows:**

1.  Introduction: the review of the dust aerosol's climate effect and the NAO's impacts on the regional climate anomalies are a little repetitive and lengthy. Address and summary the major research progress of the present work.

**Response:**

Thanks for your comment and suggestion.

We have refined the introduction, and deleted some repetitive and lengthy parts in the revised manuscript.

2.  NAO index: There are lots of definitions of the NAOI, what is the advantage of the NAOI used, and whether the result is robust if other definitions are used. The authors need to further compare the different NAOI, and clarify whether the result is subject to the NAOI.

**Response:**

Thanks for your comment and suggestion.

A systematic comparison of NAO indices by Li and Wang (2003), including the NAOI used in the manuscript, shows that the NAOI used provides a much more faithful and optimal representation of the spatiotemporal variability associated with the NAO, suggesting the NAOI as a better choice for describing and monitoring variability of the broad-scale NAO and for diagnosing relationships between the NAO and global climate variations.

We also use the NAOI provided by Climate Prediction Center, which has been used in many studies (Zuo et al., 2015; Li et al., 2021; Yao et al., 2022), for correlation analysis with the NAOI used in this manuscript. A good agreement with a correlation coefficient of 0.83 is shown between these two indices (Figure R1). The robustness of the results will not be affected by the NAOI verified by above process (Lines 595-599 in the revised manuscript).

[Figure]

**Figure R1**. The annual mean NAOI used in the manuscript (black line) and provided by Climate Prediction Center (red line) during 1980-2020.

3. The role of previous winter NAO on the following spring dust aerosols is discussed, however, as reported ENSO shows a significant role in determining the winter and spring climate anomalies over eastern China. it is of interest to further discuss whether the impacts of NAO on the aerosols are independent of ENSO.

**Response:**

Thanks for your comment and suggestion.

We have adopted the comment of the review by further examining the possible effect of ENSO on the relationship between the NAO and dust aerosols. Through partial correlation (after removing the impact of ENSO), it can be found that there is still a good correlation between the spring dust aerosols in North China and previous winter NAOI during 1980-2020 (Figure R2). Therefore, it is proved that the impact of NAO on the aerosols are independent of that of ENSO.

[Figure]

**Figure R2**. (a) Correlations between the dust aerosols during springtime and NAOI in previous winter during 1980-2020. (b) As in (a), but for the partial correlations after removing the effects of

ENSO. The black box indicates the North China. The black dotted areas are statistically significant at the 95% confidence level.

4. L218: prior to rather than previous to

**Response:**

    Thanks for your suggestion.

    We have revised it (Line 192 in the revised manuscript), as well as checked the whole manuscript and revised similar errors.

5. Line 34: The descriptions are confusing.

**Response:**

    Thanks for your comment.

    Transient eddy fluctuations contribute to the maintenance of the atmospheric energy balance through energy transport, and the energy transport process causes the divergence and convergence of energy and mass between different regions, thus forming new regions of forcing within the atmosphere (Li et al., 2019). When the transient eddy flux transport characteristics of the upstream regions of North China including the dust aerosol source regions and the Ural Ridge have disturbed, the upper-level wind speed over North China shows a weakened state.

    Therefore, it can be considered that the divergence of energy and mass have been changed over North China, which in turn has affected the wind speed over North China. To avoid confusion, we have revised the description, as shown in Lines 32-35 in the revised manuscript.

6. In the upper troposphere, it is emphasized that both the jet stream and downward transmission of high-altitude wind speed momentum contribute to the formation of the dust aerosols, however, without a detailed physical explanation of the combined role the two.

**Response:**

    Thanks for your comment.

Before the outbreak of dust aerosol events, under the impact of the jet stream, there are anomalous positive zonal winds and negative zonal winds controlling the dust source areas of Xinjiang, Mongolia and North China, respectively. Through the effect of vertical circulation, abnormal winds at high altitudes can influence middle and low altitude winds through momentum compensation and downward (Li et al., 2015; Wu et al., 2016).

Therefore, which may lead to the generation of windy weather near the dust source areas and the maintaining the dust aerosol concentration in North China, both of which contribute to the occurrence of dust aerosol events in North China. To illustrate a detailed physical explanation, we have rewritten this section, as shown in Lines 331-337 in the revised manuscript.

7. In the middle troposphere, the evolution of the trough-ridge system along with the occurrence of the dust aerosol events should be further analyzed. The involved physical process, particularly, which variable is more important in the process warranty further illustration.

**Response:**

Thanks for your comment.

Before the outbreak of dust aerosol events, the trough-ridge situation is characterized by two troughs and Ural Ridge throughout the middle-high latitudes of the Eurasian continent. The troughs are manifested as negative variations, while the Ural Ridge is manifested as gradual enhancement. On account of the strengthening advancement of the trough-ridge situation, the middle-high latitudes are dominated by the strong meridional circulation, the transport of northern cold air increases, and the southward invasion of the cold air enhances the local surface wind speed in the dust aerosol source areas, leading to the uplift of dust aerosols.

We have revised this section to show the evolution of the above circulation trend has promoted the outbreak of dust aerosol events in North China, as well as the

important role of both troughs and Ural Ridge in the process (Lines 374-377 in the revised manuscript).

8. There are lots of clerical errors, i.e.,

   L306, should be 0.1 in and 0.2 in

   L351, should be 35 °-50 °N, 70 °-110 °E

   L356-360, lengthy and repetition

   L377, should be prior to

   L404, should be 30 °-60 °N, 105 °-130 °E

   The authors should carefully check the whole manuscript.

**Response:**

Thanks for your comments and suggestions.

We have checked the whole manuscript and revised all errors.

9. Figures 2, 4, 7, 10, 13, without clarification of the axis, Figure 3, without units.

**Response:**

Thanks for the comment and suggestion.

We have revised all the legends of these figures.

**References:**

Li, J. P. and Wang, J. X. L.: A new North Atlantic Oscillation index and its variability, Adv. Atmos. Sci., 20, 661-676, https://doi.org/10.1007/BF02915394, 2003.

Li, M. Y., Yao, Y., Simmonds, I., Luo, D. H., Zhong, L. H., and Pei, L.: Linkages between the atmospheric transmission originating from the North Atlantic Oscillation and persistent winter haze over Beijing, Atmospheric Chemistry and Physics, 21, 18573-18588, https://doi.org/10.5194/acp-21-18573-2021, 2021.

Li, X., Liu, X. D.: Relation of Spring Dust-Storm Activities in Northern China and Changes of Upper Westerlies, Plateau. Meteorology (in Chinese)., 34, 1292-1300, https://doi.org/10.7522/j.issn.1000-0534.2014.00067, 2015.

Li, Y., Zhang, J. Y., Lu, Y., Zhu, J. L., and Feng, J.: Characteristics of Transient Eddy Fluxes during Blocking Highs Associated with Two Cold Events in China, Atmosphere, 10, 15, https://doi.org/10.3390/atmos10050235, 2019.

Wu, J., Kurosaki, Y., Shinoda, M., and Kai, K. J.: Regional Characteristics of Recent Dust Occurrence and Its Controlling Factors in East Asia, Sola, 12, 187-191, https://doi.org/10.2151/sola.2016-038, 2016.

Yao, Y., Zhang, W. Q., Luo, D. H., Zhong, L. H., and Pei, L.: Seasonal Cumulative Effect of Ural Blocking Episodes on the Frequent Cold events in China during the Early Winter of 2020/21, Adv. Atmos. Sci., 39, 609-624, https://doi.org/10.1007/s00376-021-1100-4, 2022.

Zuo, J. Q., Ren, H. L., and Li, W. J.: Contrasting Impacts of the Arctic Oscillation on Surface Air Temperature Anomalies in Southern China between Early and Middle-to-Late Winter, J. Clim., 28, 4015-4026, https://doi.org/10.1175/jcli-d-14-00687.1, 2015.

---

## Author Comment (AC2)

**Response to Comments of Reviewer 2**

**Manuscript number**: egusphere-2023-52

**Author(s)**: Yan Li, Falei Xu, Juan Feng, Mengying Du, Wenjun Song, Chao Li, and Wenjing Zhao

**Title**: Influence of the previous North Atlantic Oscillation (NAO) on the spring dust aerosols over North China

**General comments:**

This study investigated the relationship between the boreal winter North Atlantic Oscillation (NAO) and the spring dust events (DEs) over North China (30-40 °N, 105-120 °E) during 1980-2020. The authors demonstrated that there is a significant negative relationship between the boreal winter NAO index and the late spring DAs in the North China. The synoptic cause of such relation is characterized as the changes in the tropospheric synoptic situation in the Ural Mountains, DAs source regions in China, and North China during the DEs under the modulation of the NAO negative signal. Further, evolution mechanism of abnormal atmospheric circulation affected by the NAO negative signal is explained by the transient eddy fluxes transport, thermal wind principle, and the quasi-geostrophic theory.

This paper is very meaningful, and a great quantity of previous work has been summarized and cited. The results of this study are interesting and useful to deepen our understanding of formation mechanism of the DEs in east Asia. I would like to recommend this manuscript should be accepted subject to `minor revision`, for this study fit well with the scope of Atmospheric Chemistry and Physics.

**Response:**

Thanks to the reviewer for the helpful comments and suggestions. We have revised the manuscript seriously and carefully according to the reviewer's comments and suggestions. The point-to-point responses to the comments are listed as follows.

**Specific comments are as follows:**

1. Line 85: "it is of important practical". I suggest deleting "of" in the sentence.

**Response:**

    Yes, done.

2. Line 155,450,471: "mechanisms". I suggest revising it as "mechanism" and pay more attention to tenses of verbs.

**Response:**

    Thanks for your suggestion.

    We have revised all inappropriate uses (Line 132, 414, 431) and checked the tenses of verbs in the revised manuscript.

3. Line 140: "the DEs in northern China regions". I suggest deleting "regions".

**Response:**

    Yes, done.

4. Line 192-195: The full name of the ERA5 reanalysis datasets is not correct.

**Response:**

    Thanks for your comment.

    We have revised this description as "The atmospheric reanalysis data set, including the wind field, geopotential height field, sea level pressure field, temperature field, and vertical velocity field, obtained from the European Center for Medium-Range Weather Forecasts (ECMWF) is the fifth-generation reanalysis global atmosphere (ERA5) data set over the period 1980-2020 (horizontal resolution: $0.25°$ x $0.25°$). Compared to its predecessor, ERA-Interim, ERA5 has a modified data assimilation system and improved physical model to achieve reanalysis data information with improved quality (Hersbach et al., 2020)", as shown in Lines 166-172.

5. Line 208-210: What is the exact definition of selection criteria for the NAO abnormal years? Please describe it specifically.

**Response:**

Thanks for your suggestion.

The selection criteria for the NAO abnormal years is based on the NAOI index averaged over the winter months, then the index is normalized, and the years with a NAOI exceeding 0.5 standard deviations are recorded as NAO anomalous years. The 0.5 standard deviation is used as the criteria for the selection of abnormal years, in order to not filter excessively while retaining the NAO signal to the maximum extent possible.

According to your suggestion, a specific description has been added in Lines 181-184.

6. Line 259: "in DJF, JFM, and FMA in the early period". I suggest revising it as "in previous DJF, JFM, and FMA".

**Response:**

Yes, done.

7. Line 278: Figure 3, units needed.

**Response:**

Thanks for your suggestion.

The unit has been added to the legend in Figure 3.

8. Line 310-311: Information of the 27 DEs seems not enough. It is suggested to list 27 DEs and the years in which these DEs occurred in a table, which may be easier for readers to understand better.

**Response:**

Thanks for your comment and suggestion.

The table containing 27 DEs and the years in which these DEs occurred is seen in Table R1 and included in the revised manuscript as suggested (Table 1).

**Table R1**. Based on the CNMC selection criteria, 27 spring DEs and the years in which these DEs occurred in North China during 1980-2020.

| Years | DEs |
|---|---|
| 1980, 1981, 1982, 1983, 1984, 1985 | 19800419, 19810308, 19810325, 19810502, 19820408, 19820502, 19820508, 19830316, 19830401, 19830428, 19840301, 19840420, 19840428, 19850403 |
| 1987, 1988, 1990, 1992, 1993, 1995 | 19870317, 19880411, 19880417, 19900407, 19920411, 19930424, 19950311 |
| 1998, 2000, 2002, 2010, 2013 | 19980416, 20000327, 20000409, 20020320, 20100320, 20130309 |

9. Line 316-321: It is not clear how the 9 DEs are selected from the climate scale and the weather scale. It is better to be described in detail by means of figures or tables.

**Response:**

Thanks for your comment and suggestion.

According to your suggestion, we have illustrated the Figure explaining the DEs selected from the comprehensive consideration of climate and weather scales in the revised manuscript (Figure R1) and included in the revised manuscript as suggested (Figure 5).

[Figure]

**Figure R1**. (a) The standardized inter-annual variability of NAOI in previous winter of the years in which these DEs occurred, (b) the number of days when the value of NAOI is less than -0.5 in previous winter of the years in which these DEs occurred.

10. Line 404: Should be (30°-60°N, 105-130°E).

**Response:**

Yes, done.

11. Line 579,603: The citation of references (Gary, 2011) should be revised as (Gary, 2012).

**Response:**

Thanks for your comment and suggestion.

This error has been revised as shown in Line 534, 558 in the revised manuscript.

12. Line 728-730: The author also mentioned that both NAO and ENSO play important role in the occurrence and development of DEs in China, and the research on such synergistic effects is relatively little yet. Have you got some preliminary results? Or the authors could conduct research on this issue in the next step of work.

**Response:**

Thanks for your comment and suggestion.

From the analysis of previous work, the ENSO plays an important role in the development process of DEs in China (Li et al., 2021; Yang et al., 2022), and ENSO will also have a certain impact on the development of NAO (Graf and Zanchettin, 2012; Zhang et al., 2015). Through partial correlation (after removing the impact of ENSO), it is found that there is still a good correlation between the spring dust aerosols in North China and previous winter NAOI during 1980-2020 (Figure R2), indicating that the impact of NAO on the aerosols are independent of ENSO.

Anyway, we have not conducted systematic research on the effects of ENSO and NAO on dust aerosols. We will conduct this issue according to your suggestion in our future work.

[Figure]

**Figure R2**. Correlations between the dust aerosols during springtime and NAOI in previous winter (a) during 1980-2020. (b) As in (a), but for the partial correlations (after removal of ENSO). The black box indicates the North China. The black dotted areas are statistically significant at the 95% confidence level.

13. The labels in some figures (Figure 7, Figure 10-13) should be further enlarged, since they are rather small to distinguish.

**Response:**

Thanks for your suggestion.

We have revised these problems in all the figures.

**References:**

Graf, H.-F. and Zanchettin, D.: Central Pacific El Niño, the "subtropical bridge," and Eurasian climate, Journal of Geophysical Research: Atmospheres, 117, https://doi.org/10.1029/2011JD016493, 2012.

Hersbach, H., Bell, B., Berrisford, P., Hirahara, S., Horanyi, A., Munoz-Sabater, J., Nicolas, J., Peubey, C., Radu, R., Schepers, D., Simmons, A., Soci, C., Abdalla, S., Abellan, X., Balsamo, G., Bechtold, P., Biavati, G., Bidlot, J., Bonavita, M., De Chiara, G., Dahlgren, P., Dee, D., Diamantakis, M., Dragani, R., Flemming, J., Forbes, R., Fuentes, M., Geer, A., Haimberger, L., Healy, S., Hogan, R. J., Holm, E., Janiskova, M., Keeley, S., Laloyaux, P., Lopez, P., Lupu, C., Radnoti, G., de Rosnay, P., Rozum, I., Vamborg, F., Villaume, S., and Thepaut, J. N.: The ERA5 global reanalysis, Q. J. R. Meteorol. Soc., 146, 1999-2049, https://doi.org/10.1002/qj.3803, 2020.

Li, J., Garshick, E., Huang, S. D., and Koutrakis, P.: Impacts of El Nino-Southern Oscillation on surface dust levels across the world during 1982-2019, Sci. Total Environ., 769, 7, https://doi.org/10.1016/j.scitotenv.2020.144566, 2021.

Yang, Y., Zeng, L., Wang, H., Wang, P., and Liao, H.: Dust pollution in China affected by different spatial and temporal types of El Niño, Atmospheric Chemistry and Physics, 22, 14489-14502, https://doi.org/10.5194/acp-22-14489-2022, 2022.

Zhang, W. J., Wang, L., Xiang, B. Q., Qi, L., and He, J. H.: Impacts of two types of La Nina on the NAO during boreal winter, Clim. Dyn., 44, 1351-1366, https://doi.org/10.1007/s00382-014-2155-z, 2015.

---

## Author Comment (AC3)

**Response to Comments of Reviewer 3**

**Manuscript number**: egusphere-2023-52

**Author(s)**: Yan Li, Falei Xu, Juan Feng, Mengying Du, Wenjun Song, Chao Li, and Wenjing Zhao

**Title**: Influence of the previous North Atlantic Oscillation (NAO) on the spring dust aerosols over North China

**General comments:**

This study investigates the impacts of North Atlantic Oscillation (NAO) on spring dust aerosols over North China based on station observation data and multi reanalysis datasets. They found that late spring dust aerosols are negatively correlated with the boreal winter NAO index. They further illustrate that the changes in transient eddy momentum over the dust source region the Ural Mountains could explain the relative high spring dust levels following the negative winter NAO events. The topic is interesting and method is sound. I recommend it can be published after addressing my minor comments below.

**Response:**

Thanks to the reviewer for the helpful comments and suggestions. We have revised the manuscript seriously and carefully according to the reviewer's comments and suggestions. The point-to-point responses to the comments are listed as follows.

**Specific comments are as follows:**

1. My main comment is about how is the station observation of dust events and reanalysis data connected. Many results of this study are based on MERRA-2 dust column mass. However, how the MERRA-2 captures the dust events over the 41-years, especially those related to the negative NAO, comparing to the surface observation should be examined. Also, the dust events are separated into dust storm, blowing dust and floating dust. Are they show the same relationship with NAO?

**Response:**

Thanks for your comment and suggestion.

From the results of previous studies, MERRA-2 reanalysis data has good accuracy and applicability for studying the evolutionary situation of dust events in Asia, as well as its analysis results are more excellent compared with MODIS, OMPS, CALIPSO and Hamawari-8 data (Kang et al., 2016; Wang et al., 2018; Yao et al., 2020; Wang et al., 2021). And the frequency of dusty weather from the station data is calculated and converted into the intensity index (Wang et al., 2008), which can be used to represent the dust aerosol content. It is found that use of the intensity index and the dust column mass density from MERRA-2 reanalysis data to analyze the spatiotemporal distribution characteristics and the changing situation of dust aerosol content in China are basically the same (Figure R1).

The dust storm, blowing dust and floating dust are also closely related to the NAO, similarly to that of the dust events in our study. We select the years which that spring blowing dust and floating dust events (no dust storm events) in North China during the period of 1980-2020 occurred (Table R1).

And it is found that the previous winter sea level pressure of the North Atlantic in the years which different dust events occurred, the dipole structure of "+ -" can be observed in the lower, middle and upper troposphere (Figure R2), which is a significant negative phase characteristic of NAO, indicating that there is a significant negative correlation between the previous NAO and different types of dust events.

[Figure]

**Figure R1**. (a-d) Seasonal distribution of DI from station data, (e-h) As in (a-d), but for dust column mass density from MERRA-2 reanalysis data (The unit in (a-d) is mg m$^{-2}$).

**Table R1**. The years in which spring dust storm, blowing dust and floating dust events in North China during 1980-2020 occurred.

| Dust storm events | Blowing dust events | Floating dust events |
|---|---|---|
| None | 1980, 1981, 1982, 1983, 1984, 1988, 1990, 1995, 1998, 2000, 2002, 2010, 2013 | 1980, 1981, 1982, 1983, 1984, 1985, 1987, 1988, 1990, 1992, 1993, 1995, 1998, 2002, 2010 |

[Figure]

**Figure R2**. (a-c) Spatial distribution of the 200 hPa, 500 hPa, and 850 hPa geopotential height anomalies in previous winter of the year when the blowing dust occurred, (d-f) As in (a-c), but for the year when the flowing dust occurred.

2. In the abstract, the correlation coefficient between the boreal winter NAO index and the late spring DAs in the North China is -0.39, but I did not find thin value in the main text (probably in line 260).

**Response:**

Thanks for your comment.

We have revised the description as "Furthermore, significant correlation coefficients can be found between the spring DAs in North China and the NAOI in previous DJF, JFM, and FMA, and the correlation coefficients are -0.39, -0.40, -0.40, -0.28, respectively", as shown in Lines 226-228.

3. Line 54: Studies also reported that the dust can interact with winds in China through its radiative effect, which further weakens winds during weak wind years in winter over North China (e.g., Yang et al., 2017).

**Response:**

Thanks for your suggestion.

Study of Yang et al. (2017) is truly related to our work. We have quoted the work in the introduction section as "In particular, the radiative forcing of DAs is comparable to that of clouds on a regional-scale and has a key impact on the local weather-climate (Kaufman et al., 2002; Huang et al., 2014a; Yang et al., 2017)", as shown in Lines 40-43.

4. The authors explain the impacts of NAO on dust in North China through changes in energy and large-scale circulation. But they also showed that NAO can cause unusual rainy climate, which can influence the wet removal of dust. How this effect considered in the study.

**Response:**

Thanks for your comment and suggestion.

Previous researches have indicated that the NAO stimulates the Rossby wave and brings water vapor to the Yangtze River, China, resulting in the excessive precipitation there (Han and Zhang, 2022). In addition, the NAO is significantly correlated with the variations of East China rainfall through the Eurasian teleconnection (Wang et al., 2018). In this work, we demonstrates that during the DEs under the modulation of the NAO negative signal, North China is controlled by an anticyclonic system dominated by sinking airflow. Therefore, poor precipitation conditions lead to the weak wet-deposition of DAs during transport (Kutiel and Furman, 2003).

According to your suggestion, we have added description as shown in in Lines 388-391.

5. The mechanism of the NAO impacts may not be only limited to dust in China, but also dust over the central Asia and North Africa. The authors could add some discussion about it.

**Response:**

Thanks for your comment and suggestion.

In the revised manuscript, according to your suggestion, we added the description "Furthermore, NAO is closely related to dust activities in many regions, except for China. For example, Moulin et al. (1997) and Ginoux et al. (2004) both indicated a strong correlation between the NAO and dust activity in North Africa, using satellite data and dust transport models, respectively. Banerjee et al. (2021) and Li et al. (2022) emphasized the important role of NAO in the dust activities of South Asia and Central Asia. Hence, future studies are needed to explore the relationship between the dust activities in other regions and NAO" in discussion (Lines 688-693 in the revised manuscript).

**References:**

Banerjee, P., Satheesh, S. K., and Moorthy, K. K.: Is the Atlantic Ocean driving the recent variability in South Asian dust? Atmospheric Chemistry and Physics, 21, 17665-17685, https://doi.org/10.5194/acp-21-17665-2021, 2021.

Li, Y., Song, Y. G., Kaskaoutis, D. G., Zhang, X. X., Chen, X. L., Shukurov, N., and Orozbaev, R.: Atmospheric dust dynamics over Central Asia: A perspective view from loess deposits, Gondwana Res., 109, 150-165, https://doi.org/10.1016/j.gr.2022.04.019, 2022.

Ginoux, P., Prospero, J. M., Torres, O., and Chin, M.: Long-term simulation of global dust distribution with the GOCART model: correlation with North Atlantic Oscillation, Environ. Modell. Softw., 19, 113-128, https://doi.org/10.1016/s1364-8152(03)00114-2, 2004.

Han, J. P. and Zhang, R. H.: Influence of preceding North Atlantic Oscillation on the spring precipitation in the middle and lower reaches of the Yangtze River valley, Int. J. Climatol., 42, 4728-4739, https://doi.org/10.1002/joc.7500, 2022.

Kang, L. T., Huang, J. P., Chen, S. Y., and Wang, X.: Long-term trends of dust events over Tibetan Plateau during 1961-2010, Atmos. Environ., 125, 188-198, https://doi.org/10.1016/j.atmosenv.2015.10.085, 2016.

Kurosaki, Y. and Mikami, M.: Recent frequent dust events and their relation to surface wind in East Asia, Geophysical Research Letters, 30, 4, https://doi.org/10.1029/2003gl017261, 2003.

Moulin, C., Lambert, C. E., Dulac, F., and Dayan, U.: Control of atmospheric export of dust from North Africa by the North Atlantic oscillation, Nature, 387, 691-694, https://doi.org/10.1038/42679, 1997.

Yao, W. R., Che, H. Z., Gui, K., Wang, Y. Q., and Zhang, X. Y.: Can MERRA-2 Reanalysis Data Reproduce the Three-Dimensional Evolution Characteristics of a Typical Dust Process in East Asia? A Case Study of the Dust Event in May 2017, Remote Sens., 12, 18, https://doi.org/10.3390/rs12060902, 2020.

Wang, X., Huang, J. P., Ji, M. X., and Higuchi, K.: Variability of East Asia dust events and their long-term trend, Atmos. Environ., 42, 3156-3165, https://doi.org/10.1016/j.atmosenv.2007.07.046, 2008.

Wang, T. H., Tang, J. Y., Sun, M. X., Liu, X. W., Huang, Y. X., Huang, J. P., Han, Y., Cheng, Y. F., Huang, Z. W., and Li, J. M.: Identifying a transport mechanism of dust aerosols over South Asia to the Tibetan Plateau: A case study, Sci. Total Environ., 758, 11, https://doi.org/10.1016/j.scitotenv.2020.143714, 2021.

Wang, Z. Q., Yang, S., Lau, N. C., and Duan, A. M.: Teleconnection between Summer NAO and East China Rainfall Variations: A Bridge Effect of the Tibetan Plateau, J. Clim., 31, 6433-6444, https://doi.org/10.1175/jcli-d-17-0413.1, 2018.

Wang, Z. Q., Yang, S., Lau, N. C., and Duan, A. M.: Teleconnection between Summer NAO and East China Rainfall Variations: A Bridge Effect of the Tibetan Plateau, J. Clim., 31, 6433-6444, https://doi.org/10.1175/jcli-d-17-0413.1, 2018.